# Specified-duration shapers for suppressing residual vibrations

**Brian Byunghyun Kang** *

School of Intelligent Mechatronics Engineering, Sejong University, Seoul, Korea

* brianbkang@sejong.ac.kr

## Abstract

Input-shaping control has received considerable research attention for suppressing residual vibrations. Although numerous studies have been conducted on designing input shapers with arbitrary robustness to modeling errors, no studies have focused on the design of input shapers with arbitrarily specified shaping times. In this study, a specified-duration (SD) shaper, which is an input shaper with an arbitrarily specified shaping time, and a systematic method to design an SD shaper using impulse vectors are proposed. As the specified shaping time increases, the SD shaper increases the number of impulses one by one according to the number of added derivative constraints, thereby improving robustness to modeling errors. The performance of the SD shaper was evaluated for a second-order system through computer simulations. The simulation results revealed that the SD shaper suppresses residual vibrations of the vibratory system at the specified shaping time. The validity of the SD shaper was experimentally verified using a horizontal beam vibration apparatus. The results of this study provide insight into the development of vibration suppression strategies with input shaping control.

## Introduction

In engineering systems, it is often beneficial to be able to rapidly reduce vibrations to avoid system performance degradation. For example, the forced lateral vibration of a building during earthquakes should be minimized to avoid structural damage [1]. Another example is that when cranes [2–4], disk drives [5, 6], sloshing liquid containers [7–10], industrial robots [11], and soft robots [12, 13] are operated, residual vibrations should be mitigated swiftly to reduce working hours and increase working efficiency.

Input-shaping control has attracted considerable research attention for removing residual vibration. Input-shaping control was first introduced as posicast control in the 1950s by O. J. M. Smith [14]. Since the 1980s, N. Singer, J. Hyde, W. Singhose, and T. Singh have considerably improved input-shaping control by adding derivative constraints [15–19]. This control exhibits considerable residual vibration removal efficiency without an additional feedback loop in a vibratory system.

The block diagram of a general input-shaping control system is displayed in Fig 1. The vibratory system in Fig 1 includes a feedback loop and a feedback controller. Input-shaping

**Data Availability Statement:** https://github.com/BBKangIRL/SDshaperExp.git.

**Funding:** This work was supported by the National Research Foundation of Korea (NRF) grant funded by the Korea government (MSIT) (No.

2021R1G1A109521912) and the faculty research fund of Sejong University in 2021.

**Competing interests:** The authors have declared that no competing interests exist.

control does not affect the stability of the vibratory system because the input shaper is connected in series outside the feedback loop as shown in Fig 1. The input shaper consists of a sequence of impulse functions. The convolution of this impulse sequence (i.e., input shaper) with the command input of the system can remove residual vibration caused by the command input [20].

Kang [21] has recently introduced an impulse vector as a mathematical tool for designing an input shaper. Residual vibrations can be suppressed regardless of the damping condition by designing input shapers using impulse vectors such that the sum of the impulse vectors in the polar coordinate plane becomes zero.

In the 1990s, specified-insensitivity (SI) shapers were developed to be robust to modeling errors with arbitrarily specified insensitivity [22–25]. The SI shaper was designed to ensure that the sensitivity function $V$ of the input shaper was within a tolerance value $V_{tol}$ for a specified range of modeling errors. The sensitivity function $V$ indicates the relative magnitude of residual vibration for modeling errors in natural frequency and damping ratio. Thus, to design an SI shaper with, for example, $V_{tol} = 0.05$, we first specify an arbitrary 5% insensitivity $I_{0.05}$, which forms the range of modeling errors such that the relative residual vibration is within 5%, and then design the input shaper to satisfy this $I_{0.05}$ condition. The 5% insensitivity $I_{0.05}$ represents the robustness to modeling errors of the input shaper when $V_{tol} = 0.05$.

The shaping time (or shaper duration) as well as the robustness to modeling errors are important factors in the design of input shapers. Because the shaping time represents the time to remove residual vibrations, smaller shaping times are preferred. As the robustness to modeling errors (e.g., 5% insensitivity) increases, the shaping time also increases nonlinearly in general [26]. Moreover, the shaping time can be reduced to a certain limit because the smaller the shaping time is, the greater the actuator power is required. Although the shaping time is an important factor in the design of input shapers [27], no studies have been conducted on designing an input shaper that can satisfy an arbitrarily specified shaping time.

In this paper, a novel specified-duration (SD) shaper that satisfies an arbitrarily specified shaping time with the largest 5% insensitivity to modeling errors when the magnitude of the last impulse is taken as a free parameter is proposed. The SD shaper is an input shaper that is designed to remove residual vibration during a given specified shaping time. As the specified shaping time increases, the SD shaper increases the number of impulses one by one according to the number of added derivative constraints, thereby improving robustness to modeling errors. A systematic method is presented to design the SD shaper with an arbitrarily specified shaping time by using impulse vectors. The performance of the proposed SD shaper is

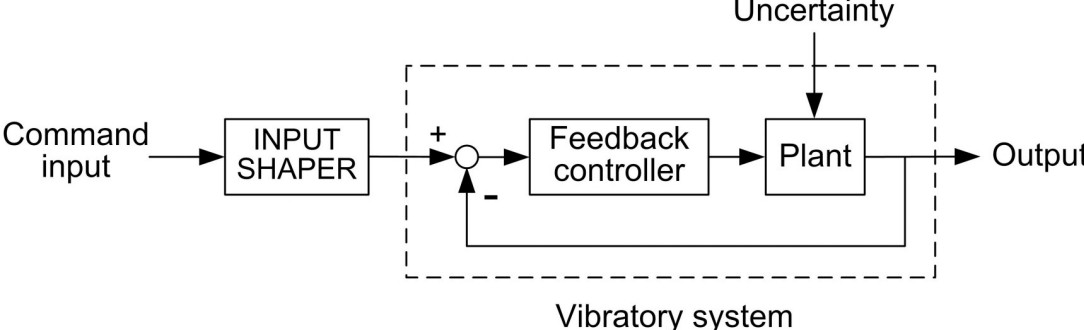

**Fig 1. Block diagram of an input-shaping control system.** The input shaper is connected in series with a vibratory system and located outside feedback loops. Thus, the shaper does not affect the stability of the vibratory system.

evaluated through computer simulations. Furthermore, the performance of the SD shaper is verified experimentally using a horizontal beam vibration apparatus.

This paper is organized as follows. Section 2 presents a systematic method to design an SD shaper with an arbitrarily shaping time by using impulse vectors. In Section 3, the performance of the SD shaper is discussed via MATLAB simulations using sensitivity curves and step responses. Section 4 demonstrates the validity of the SD shaper experimentally. The conclusion of the paper is presented in Section 5.

## Design of an SD shaper

In this section, the design of an SD shaper with an arbitrarily specified shaping time $t_s$ is described. Vibratory system dynamics is assumed to be an undamped or underdamped second-order system $\omega_n^2/(s^2 + 2\zeta\omega_n s + \omega_n^2)$, where $\omega_n$ is the undamped natural frequency and $\zeta$ is the damping ratio of the system. An impulse vector $\mathbf{I}_i$ ($i = 1, 2, \cdots, N$) with magnitude $I_i$ and angle $\theta_i$ in the polar coordinate plane was proposed by Kang [21], which was as follows:

$$I_i = A_i\, e^{\zeta\omega_n t_i},\ i = 1, 2, \cdots, N \tag{1}$$

$$\theta_i = \omega_d\, t_i,\ i = 1, 2, \cdots, N \tag{2}$$

where $A_i$ is the impulse magnitude of the $i^{\text{th}}$ impulse function, $t_i$ is the impulse time of the $i^{\text{th}}$ impulse function, and $\omega_d$ is the damped natural frequency of the system, which is defined as

$$\omega_d = \omega_n\sqrt{1 - \zeta^2}. \tag{3}$$

We can design an SD shaper with an arbitrarily specified shaping time $t_s$ by using the impulse vector. The shaping time $t_s$ is equal to the last impulse time $t_N$ of the impulse sequence in which $N$ represents the number of impulses of the input shaper. First, for an arbitrarily specified shaping time $t_s = t_N$, the angle $\theta_N$ of the last impulse vector $\mathbf{I}_N$ is determined using the impulse vector definition, Eq (2), and this angle is fixed for the SD shaper design.

$$\theta_N = \omega_d t_s \tag{4}$$

Thus, the given shaping time $t_s$ determines the angle $\theta_N$ of $\mathbf{I}_N$, and the magnitude $I_N$ of $\mathbf{I}_N$ determines $A_N$. Here, $I_N$ is a value between 0 and $e^{\zeta\omega_n t_s}$ because $0 < A_N < 1$.

The damped period $T_d$ of the second-order system with a damping ratio $\zeta$ is expressed as follows:

$$T_d = \frac{2\pi}{\omega_d}. \tag{5}$$

The dimensionless shaping time $T_s$ is defined as $t_s$ divided by the damped period $T_d$.

$$T_s = \frac{t_s}{T_d} \tag{6}$$

Then, the actual shaping time $t_s$ can be expressed as $t_s = T_d T_s$ and the angle of the final impulse vector $\mathbf{I}_N$ of the SD shaper can be expressed as $\theta_N = \omega_d T_d T_s$.

It is well known that the ZV, ZVD and $\text{ZVD}^n$ shapers have the shaping times of a half period, one period, and $(n + 1)/2$ period, which are $T_s = 0.5$, $T_s = 1$, and $T_s = (n + 1)/2$, respectively [21]. Also, it is well known that the residual vibration is removed if the resultant of the impulse vectors in an impulse vector diagram is zero [21]. Moreover, we need some flexibility in order to achieve robustness to modeling errors of the SD shaper, which can be accomplished

by adding an impulse and letting its magnitude serve as a free parameter. Observing these facts, the rule of deciding the number of impulses of SD shapers are determined as follows.

If the dimensionless shaping time $T_s$ is between 0.5 and 1 (that is, between half and a period), the SD shaper consists of three impulses. If $T_s$ is between 1 and 1.5, the SD shaper consists of four impulses, and if it is between 1.5 and 2, it consists of five impulses, and so on. Thus, we have the following expression:

$$
\begin{aligned}
0.5 < T_s \leq 1 &\quad \rightarrow \quad N = 3 \\
1 < T_s \leq 1.5 &\quad \rightarrow \quad N = 4 \\
1.5 < T_s \leq 2 &\quad \rightarrow \quad N = 5 \\
&\quad \vdots
\end{aligned}
\tag{7}
$$

Eq (7) can be explained as follows. If $T_s$ is between 0.5 and 1, the angle of the final impulse vector is in $\pi < \theta_N \leq 2\pi$, and thus, an impulse vector diagram with zero resultant can be drawn using three impulse vectors, as shown in Fig 2. Therefore, three impulses are required. Three impulses have four unknowns $A_1, A_2, A_3, t_2$ since $t_3$ is given from the specified shaping time. These four unknowns can be obtained by simultaneously solving two residual vibration constraints and one normalization constraint with $A_3$ chosen in advance by the designer. Without losing generality, we set $t_1 = 0$. Further, if $T_s$ is between 1 and 1.5, the angle of the final impulse vector is in $2\pi < \theta_N \leq 3\pi$, and an impulse vector diagram with zero resultant can be drawn using four impulse vectors, as shown in Fig 3, which corresponds to an input shaper with four impulses. The four impulses can be obtained by simultaneously solving two residual vibration constraints, two derivative constraints, and a normalization constraint (six unknowns $A_1, A_2, A_3, A_4, t_2, t_3$ with $A_4$ chosen in advance by the designer). Similarly, if $T_s$ is

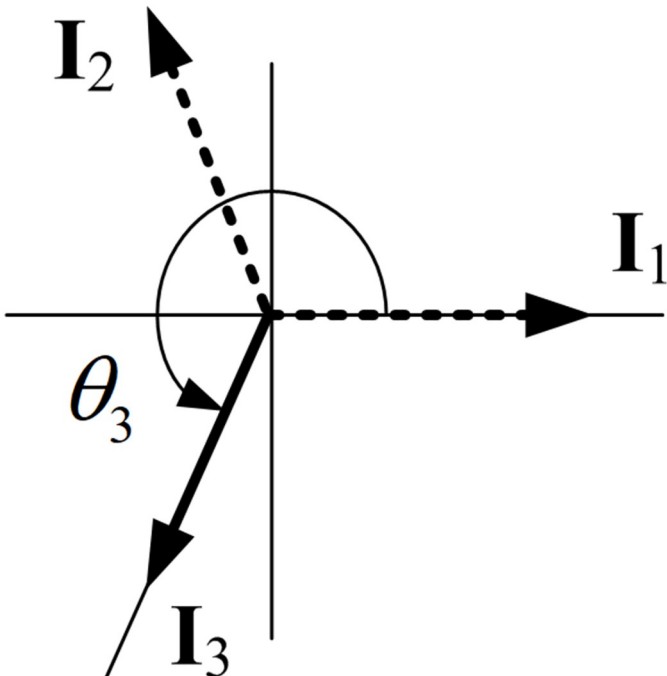

**Fig 2. Impulse vector diagram of an SD shaper with a shaping time $0.5 < T_s \leq 1$.** Because the shaping time $T_s$ is in $0.5 < T_s \leq 1$, the SD shaper is composed of three impulses.

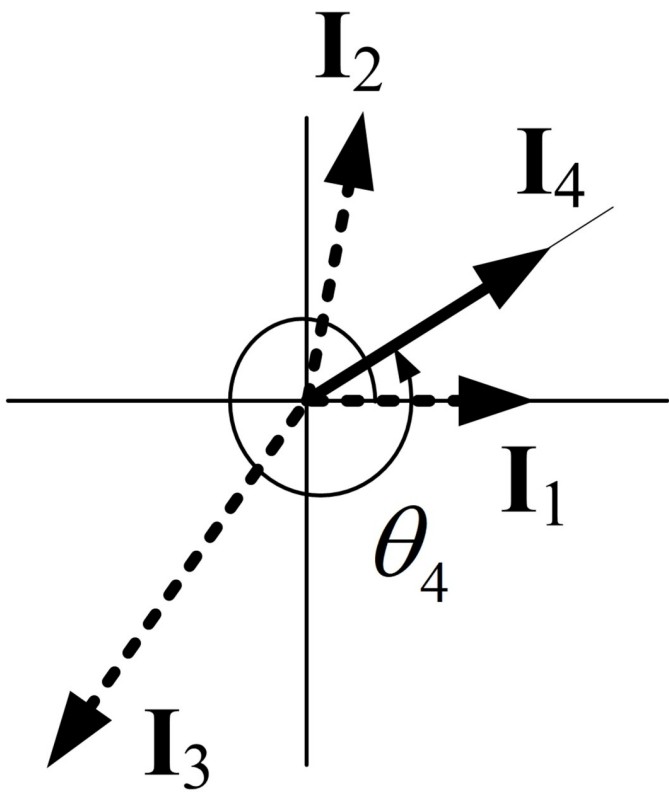

**Fig 3. Impulse vector diagram of an SD shaper with a shaping time $1 < T_s \leq 1.5$.** Because the shaping time $T_s$ is between $1 < T_s \leq 1.5$, the SD shaper is composed of four impulses.

between 1.5 and 2, Eq (7) states that we should use five impulses; these five impulses can be obtained by simultaneously solving two residual vibration constraints, two derivative constraints, two second-order derivative constraints, and a normalization constraint (eight unknowns $A_1$, $A_2$, $A_3$, $A_4$, $A_5$, $t_2$, $t_3$, $t_4$, with $A_5$ chosen in advance by the designer) [21–24]. For an example that provides a partial justification for using five impulses in this case, refer to the last paragraph of the next section, Analysis of SD shapers.

Using the final impulse magnitude $A_N$ as a free parameter to minimize the sensitivity to modeling errors, we can maximize the 5% insensitivity by increasing $A_N$ discretely by 0.01 from 0.01 to 0.99 in the design of SD shapers.

The design procedure of SD shapers can be summarized as follows:

1. Determine the number of impulses $N$ from the specified shaping time using Eq (7).

2. Obtain the constraint equations for the SD shaper with $N$ impulses.

3. Find impulse magnitudes and times $A_i$, $t_i$ by solving the constraint equations analytically or numerically, in which the last impulse magnitude $A_N$ is a free parameter (discrete values with 0.01 intervals) to maximize the 5% insensitivity to modeling errors.

Here, the design of an SD shaper with $0.5 < T_s \leq 1$ is described. If $T_s$ is between 0.5 and 1, we obtain $N = 3$, as determined by Eq (7), and the impulse vector diagram of the SD shaper can be drawn as shown in Fig 2. Here, $\theta_3$ is determined from the given shaping time $t_3$ ($= t_s$), and $\theta_2$, $I_1$, $I_2$, $I_3$ are unknowns. To ensure residual vibration $V$ is zero when there is no modeling error, the sum of impulse vectors must be zero. Therefore, the SD shaper for $0.5 < T_s \leq 1$

should satisfy the following constraints:

$$I_1 + I_2 \cos \theta_2 + I_3 \cos \theta_3 = 0 \tag{8}$$

$$I_2 \sin \theta_2 + I_3 \sin \theta_3 = 0 \tag{9}$$

$$A_1 + A_2 + A_3 = 1 \tag{10}$$

Eqs (8)–(10) have three unknowns, namely $A_1$, $A_2$ and $t_2$ because $\theta_3$, $I_3$ are given. $\theta_3$ is obtained from the specified shaping time, and $I_3$ is given from the free parameter $A_3$ that is used for maximizing the 5% insensitivity to modeling errors. Thus, Eqs (8)–(10) have a solution. Here, $I_1 = A_1$, $I_2 = A_2 e^{\zeta \omega_n t_2}$, $I_3 = A_3 e^{\zeta \omega_n t_3}$ and $\theta_2 = \omega_d t_2$. For a given $\theta_3$, we assume a specific value of the final impulse magnitude $A_3 (0 < A_3 < 1)$, and find $A_1$, $A_2$, $t_2$ that satisfy Eqs (8)–(10). Then, we vary the value of $A_3$, and find $A_1$, $A_2$, $t_2$ that satisfy Eqs (8)–(10), repeatedly.

SD shapers are obtained separately for two cases with $\zeta = 0$ and $0 < \zeta < 1$.

In case of $\zeta = 0$, Eqs (8) and (9) can be expressed as follows:

$$A_1 + A_2 \cos \theta_2 + A_3 \cos \theta_3 = 0 \tag{11}$$

$$A_2 \sin \theta_2 + A_3 \sin \theta_3 = 0 \tag{12}$$

By rearranging Eqs (10) and (11), we obtain the following expression:

$$A_2(\cos \theta_2 - 1) = A_3(1 - \cos \theta_3) - 1 \tag{13}$$

and, from Eq (12), we have the following expression:

$$A_2 \sqrt{1 - \cos^2 \theta_2} = -A_3 \sin \theta_3. \tag{14}$$

Dividing Eq (14) by Eq (13) results in the following expression:

$$\frac{\sqrt{1 - \cos^2 \theta_2}}{\cos \theta_2 - 1} = \frac{-A_3 \sin \theta_3}{A_3(1 - \cos \theta_3) - 1} \triangleq b. \tag{15}$$

By selecting a specific value of $A_3$ in Eq (15), $b$ is determined, and Eq (15) can be expressed as follows:

$$(b^2 + 1) \cos^2 \theta_2 - 2b^2 \cos \theta_2 + (b^2 - 1) = 0. \tag{16}$$

Eq (16) is a quadratic equation for $\cos\theta_2$. Because $\cos\theta_2 < 1$, the solution of Eq (16) is expressed as follows:

$$\cos \theta_2 = \frac{b^2 - 1}{b^2 + 1}. \tag{17}$$

Thus, the angle $\theta_2$ of impulse vector $\mathbf{I}_2$ is obtained as follows:

$$\theta_2 = \cos^{-1}\left(\frac{b^2 - 1}{b^2 + 1}\right). \tag{18}$$

When $A_3$ is specified, $\theta_2$ is determined by Eq (18). Because $\theta_3$ is given, the impulse magnitudes $A_1$, $A_2$ can be obtained as follows:

$$A_2 = \frac{A_3(1 - \cos\theta_3) - 1}{\cos\theta_2 - 1}$$

$$A_1 = 1 - A_2 - A_3$$

(19)

From Eqs (18) and (19), $A_1$, $A_2$, $\theta_2$ can be calculated repeatedly by increasing $A_3$ by 0.01 from 0.01 to 0.99, and, for each case, an input shaper $A_1\delta(t) + A_2\delta(t - t_2) + A_3\delta(t - t_3)$ can be obtained, in which $\delta(t)$ implies Dirac delta function. For each input shaper $A_1\delta(t) + A_2\delta(t - t_2) + A_3\delta(t - t_3)$, a sensitivity curve is drawn and 5% insensitivity $I_{0.05}$ is obtained. Among these sensitivity curves, the input shaper $A_1\delta(t) + A_2\delta(t - t_2) + A_3\delta(t - t_3)$ with the largest $I_{0.05}$ is selected as the SD shaper. The iterative procedure of finding numerical solutions can be performed using numeric computing platforms (e.g., MATLAB).

The sensitivity curve is a plot of the sensitivity function $V$ with respect to modeling errors $\omega_n/\hat{\omega}_n$, in which $\hat{\omega}_n$ represents the modeled value of natural frequency, and $\omega_n$ is the actual value of natural frequency. In the sensitivity curves, $\hat{\omega}_n$ is assumed to be fixed, and $\omega_n$ is a varying parameter. The sensitivity function $V$ represents the relative magnitude of residual vibration and is defined as follows [21]:

$$V = e^{-\zeta\omega_n t_N}\sqrt{C^2 + S^2}$$

(20)

where

$$C = \sum_{i=1}^{N} A_i \, e^{\zeta\,\omega_n\,t_i} \cos\omega_d t_i, \quad S = \sum_{i=1}^{N} A_i \, e^{\zeta\,\omega_n\,t_i} \sin\omega_d t_i$$

(21)

In Eqs (20) and (21), $\omega_n$, $\omega_d$, $\zeta$ are the varying actual values of the system, whereas $t_i$, $A_i$ are the constant values obtained from the fixed modeled values $\hat{\omega}_n$, $\hat{\omega}_d$, $\hat{\zeta}$. The 5% insensitivity $I_{0.05}$ is the distance between two points where $V = 0.05$ line and the sensitivity curve intersect.

As an example, let us determine an SD shaper with a shaping time $t_s = 0.3$s for an undamped system $\omega_n^2 / (s^2 + \omega_n^2)$ with a natural frequency $\omega_n$ of 2 Hz (= $4\pi$ rad/s). In this case, the dimensionless shaping time is $T_s = 0.6$ according to Eq (6). Thus, the SD shaper is composed of three impulses according to Eq (7). The SD shaper can be obtained using Eqs (1), (2), (15), (18) and (19) as follows:

$$\begin{bmatrix} t_i \\ A_i \end{bmatrix} = \begin{bmatrix} 0, & 0.1305, & 0.3 \\ 0.3484, & 0.2416, & 0.41 \end{bmatrix}$$

(22)

In this SD shaper, the angle $\theta_3$ of the final impulse vector is $1.2\pi$ rad by Eq (4).

Next, an SD shaper with $0.5 < T_s \le 1$ is obtained for an underdamped system with a damping ratio $0 < \zeta < 1$. If $0 < \zeta < 1$, we cannot determine a closed-form solution such as Eq (19) for given $A_3$ because Eqs (8) and (9) include exponential functions. However, a numerical solution for Eqs (8)–(10) can be obtained because nonlinear simultaneous equations can be solved numerically along with the impulse vector definitions using the fsolve() function in MATLAB. Next, by increasing $A_3$ by 0.01 from 0.01 to 0.99, a numerical solution can be determined for $A_1$, $A_2$ and $\theta_2$. Then, a sensitivity curve can be drawn for each input shaper. Among these sensitivity curves, the input shaper $A_1\delta(t) + A_2\delta(t - t_2) + A_3\delta(t - t_3)$ that maximizes 5% insensitivity $I_{0.05}$ is selected as the SD shaper for the system with $0 < \zeta < 1$.

As an example, let us determine an SD shaper with a shaping time $t_s = 0.3$s for an underdamped system $\omega_n^2/(s^2 + 2\zeta\omega_n s + \omega_n^2)$ with a natural frequency $\omega_n$ of 2 Hz and a damping ratio $\zeta$ of 0.1. In this case, the dimensionless shaping time is $T_s = 0.597$ according to Eq (6). Thus, the SD shaper is composed of three impulses according to Eq (7). The SD shaper obtained numerically using MATLAB is expressed as follows:

$$\begin{bmatrix} t_i \\ A_i \end{bmatrix} = \begin{bmatrix} 0, & 0.1264, & 0.3 \\ 0.4129, & 0.2442, & 0.3430 \end{bmatrix} \tag{23}$$

From Eq (4), the angle $\theta_3$ of the final impulse vector in this SD shaper is $1.194\pi$ rad.

Next, we obtain an SD shaper for the case when $T_s$ is between 1 and 1.5. Even when $T_s$ is between 1 and 1.5, an input shaper with three impulses can be obtained from constraints Eqs (8)–(10). However, to maximize 5% insensitivity, we obtain an SD shaper with four impulses as displayed in Fig 3 by adding derivative constraints. Therefore, when $T_s$ is greater than 1 and less than or equal to 1.5, the SD shaper with $N = 4$ should satisfy the following constraints:

$$\begin{aligned} &I_1 + I_2 \cos\theta_2 + I_3 \cos\theta_3 + I_4 \cos\theta_4 = 0 \\ &I_2 \sin\theta_2 + I_3 \sin\theta_3 + I_4 \sin\theta_4 = 0 \\ &I_2 t_2 \cos\theta_2 + I_3 t_3 \cos\theta_3 + I_4 t_4 \cos\theta_4 = 0 \\ &I_2 t_2 \sin\theta_2 + I_3 t_3 \sin\theta_3 + I_4 t_4 \sin\theta_4 = 0 \\ &A_1 + A_2 + A_3 + A_4 = 1 \end{aligned} \tag{24}$$

Two derivative constraints, the third and fourth equations of Eq (24), are obtained by differentiating two residual vibration constraints, the first and second equations of Eq (24), with respect to $\omega_n$. The first and second constraints of Eq (24) can be expressed as functions of $\omega_n$ as follows:

$$C(\omega_n) = A_1 + A_2 e^{\zeta\omega_n t_2} \cos(\omega_d t_2) + A_3 e^{\zeta\omega_n t_3} \cos(\omega_d t_3) + A_4 e^{\zeta\omega_n t_4} \cos(\omega_d t_4) = 0$$
$$S(\omega_n) = A_2 e^{\zeta\omega_n t_2} \sin(\omega_d t_2) + A_3 e^{\zeta\omega_n t_3} \sin(\omega_d t_3) + A_4 e^{\zeta\omega_n t_4} \sin(\omega_d t_4) = 0$$

Then, we can obtain two equations, $dC/d\omega_n = 0$ and $dS/d\omega_n = 0$, by differentiating $C(\omega_n)$ and $S(\omega_n)$ with respect to $\omega_n$. These two equations are equivalent to the third and fourth constraints of Eq (24), respectively.

Regardless of whether $\zeta$ is 0, or greater than 0 in this case, by increasing $A_4$ by 0.01 from 0.01 to 0.99, we can determine $A_1$, $A_2$, $A_3$, $t_2$, and $t_3$ numerically from Eq (24) and impulse vector definitions (1), (2). The sensitivity curve is then drawn for each input shaper with $A_1$, $A_2$, $A_3$, $t_2$, $t_3$. The input shaper $A_1\delta(t) + A_2\delta(t - t_2) + A_3\delta(t - t_3) + A_4\delta(t - t_4)$ that maximizes 5% insensitivity, is selected as the SD shaper for $1 < T_s \leq 1.5$.

As an example, let us determine an SD shaper with a shaping time $t_s = 0.6$ s for the aforementioned underdamped system $\omega_n^2/(s^2 + 2\zeta\omega_n s + \omega_n^2)$ with a natural frequency $\omega_n$ of 2 Hz, and a damping ratio $\zeta$ of 0.1. The dimensionless shaping time is $T_s = 1.194$ according to Eq (6). Thus, the SD shaper is composed of four impulses according to Eq (7). The SD shaper can be obtained numerically using MATLAB as follows:

$$\begin{bmatrix} t_i \\ A_i \end{bmatrix} = \begin{bmatrix} 0, & 0.2095, & 0.3943, & 0.6 \\ 0.2370, & 0.3701, & 0.2846, & 0.1082 \end{bmatrix} \tag{25}$$

Next, we obtain an SD shaper for the case when $T_s$ is between 1.5 and 2. When $T_s$ is between 1.5 and 2, to maximize the 5% insensitivity, we obtain the SD shaper with five impulses by

adding second-order derivative constraints. Therefore, when $T_s$ is greater than 1.5 and less than or equal to 2, the SD shaper with $N = 5$ should satisfy the following constraints:

$$
\begin{aligned}
&I_1 + I_2 \cos \theta_2 + I_3 \cos \theta_3 + I_4 \cos \theta_4 + I_5 \cos \theta_5 = 0 \\
&I_2 \sin \theta_2 + I_3 \sin \theta_3 + I_4 \sin \theta_4 + I_5 \sin \theta_5 = 0 \\
&I_2 t_2 \cos \theta_2 + I_3 t_3 \cos \theta_3 + I_4 t_4 \cos \theta_4 + I_5 t_5 \cos \theta_5 = 0 \\
&I_2 t_2 \sin \theta_2 + I_3 t_3 \sin \theta_3 + I_4 t_4 \sin \theta_4 + I_5 t_5 \sin \theta_5 = 0 \\
&I_2 t_2^2 \cos \theta_2 + I_3 t_3^2 \cos \theta_3 + I_4 t_4^2 \cos \theta_4 + I_5 t_5^2 \cos \theta_5 = 0 \\
&I_2 t_2^2 \sin \theta_2 + I_3 t_3^2 \sin \theta_3 + I_4 t_4^2 \sin \theta_4 + I_5 t_5^2 \sin \theta_5 = 0 \\
&A_1 + A_2 + A_3 + A_4 + A_5 = 1
\end{aligned}
\tag{26}
$$

Two second-order derivative constraints, the fifth and sixth equations of Eq (26), are obtained by differentiating two derivative constraints, the third and fourth equations of Eq (26), with respect to $\omega_n$. Regardless of whether $\zeta$ is 0 or greater than 0, by increasing $A_4$ by 0.01 from 0.01 to 0.99, we can determine $A_1$, $A_2$, $A_3$, $A_4$, $t_2$, $t_3$ and $t_4$ numerically from Eq (26) and impulse vector definitions (1), (2). The sensitivity curve is then drawn for each input shaper with $A_1$, $A_2$, $A_3$, $A_4$, $t_2$, $t_3$ and $t_4$. The input shaper $A_1 \delta(t) + A_2 \delta(t - t_2) + A_3 \delta(t - t_3) + A_4 \delta(t - t_4) + A_5 \delta(t - t_5)$ that maximizes 5% insensitivity is selected as the SD shaper for $1.5 < T_s \leq 2$.

As an example, let us determine an SD shaper with a shaping time $t_s = 0.85$ s for the aforementioned underdamped system $\omega_n^2 / (s^2 + 2\zeta\omega_n s + \omega_n^2)$ with a natural frequency $\omega_n$ of 2 Hz, and a damping ratio $\zeta$ of 0.1. The dimensionless shaping time is $T_s = 1.6915$ according to Eq (6). Thus, the SD shaper is composed of five impulses according to Eq (7). The SD shaper with $t_s = 0.85$ s can be obtained numerically using MATLAB as follows:

$$
\begin{bmatrix} t_i \\ A_i \end{bmatrix} = \begin{bmatrix} 0, & 0.2273, & 0.4346 & 0.6335, & 0.85 \\ 0.1436, & 0.3301, & 0.3060, & 0.1756, & 0.0447 \end{bmatrix}
\tag{27}
$$

When the dimensionless shaping time $T_s$ is greater than 2, the SD shaper can be designed by repeating the aforementioned numerical analysis procedure by increasing the number of impulses by one. However, if the shaping time of the SD shaper is longer than two natural periods, the usefulness of the SD shaper decreases because the time for removing residual vibration increases correspondingly. Therefore, SD shapers with more than two natural periods were not considered. Moreover, if the dimensionless shaping time is less than 0.5, we require negative impulses to remove residual vibrations. Thus, SD shapers with $T_s < 0.5$ were not considered in this study.

Note that an SD shaper satisfying all nonlinear constraints always exists because an impulse vector diagram satisfying all nonlinear constraints of the SD shaper can always be drawn geometrically; then, the SD shaper can be obtained from the impulse vector diagram.

## Analysis of SD shapers

The performance of the SD shaper obtained in the previous section is evaluated through sensitivity curves and unit-step responses. For the undamped system $\omega_n^2 / (s^2 + \omega_n^2)$ modeled with a natural frequency of 2 Hz (i.e., $\hat{\omega}_n = 2$ Hz), the SD shaper with a shaping time $t_s = 0.3$s is expressed by Eq (22). The sensitivity curve of the SD shaper (22) is shown in Fig 4. For comparison, the sensitivity curves of the well-known ZV (zero vibration) shaper and ZVD (zero vibration and derivative) shaper [18] are also shown in Fig 4. The robustness of the SD shaper to modeling errors shown in Fig 4 is between the ZV and ZVD shapers if the actual natural

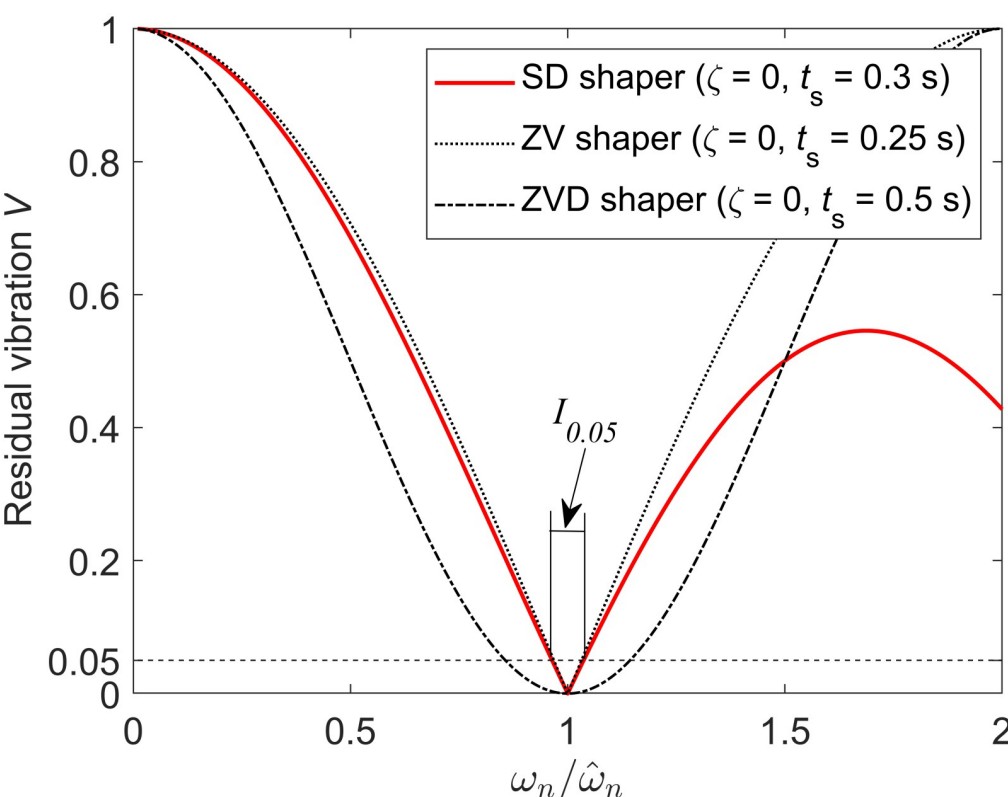

**Fig 4. Sensitivity curve of the SD shaper with the shaping time $t_s = 0.3$s for an undamped system $\omega_n^2/(s^2 + \omega_n^2)$ with modeled natural frequency $\hat{\omega}_n = 2$ Hz.** Because the dimensionless shaping time is 0.6, the SD shaper is composed of three impulses. The 5% insensitivity $I_{0.05}$ of this SD shaper is 0.073.

frequency is less than approximately 150% of the modeled value; however, the robustness of the SD shaper to modeling errors is better than that of the ZV and ZVD shapers if the actual natural frequency is 150% larger than the modeled value. The 5% insensitivity of the SD shaper is 0.073, which is between those of the ZV shaper ($I_{0.05} = 0.063$) and ZVD shaper ($I_{0.05} = 0.287$). The sensitivity curves of ZV and ZVD shapers for undamped systems are symmetric about the axis $\omega_n/\hat{\omega}_n = 1$, but the sensitivity curves of SD shapers are asymmetric even for undamped systems, as shown in Fig 4. In Fig 4, $\hat{\omega}_n$ is a fixed modeled value, and $\omega_n$ is a varying actual value.

Fig 5 shows the unit-step responses of the input-shaping control system with an SD shaper when $\zeta = 0$. Fig 5A shows the unit-step responses when no modeling error ($\omega_n = \hat{\omega}_n = 2$ Hz) exists, and Fig 5B shows the unit-step responses when 20% modeling error exists in natural frequency ($\omega_n = 2.4$ Hz, $\hat{\omega}_n = 2$ Hz). If no modeling error exists, the SD shaper has a shaping time of exactly 0.3 s as designed in Eq (22). If 20% modeling error exists in natural frequency, the magnitude of the residual vibration is between those of the ZV and ZVD shaper as shown in Fig 5B, which is consistent with the sensitivity curve trend in Fig 4.

For the underdamped system $\omega_n^2/(s^2 + 2\zeta\omega_n s + \omega_n^2)$ modeled with a natural frequency of 2 Hz and a damping ratio of 0.1, SD shapers with a shaping time of 0.3, 0.6, and 0.85 s are obtained as Eqs (23), (25) and (27), with three, four, and five impulses, respectively. The sensitivity curves for these SD shapers are shown in Fig 6, in which the robustness of SD shapers to modeling errors improves as the shaping time increases. The 5% insensitivities $I_{0.05}$ of these SD shapers calculated using MATLAB were 0.088, 0.452, and 1.133, respectively. Fig 6 shows

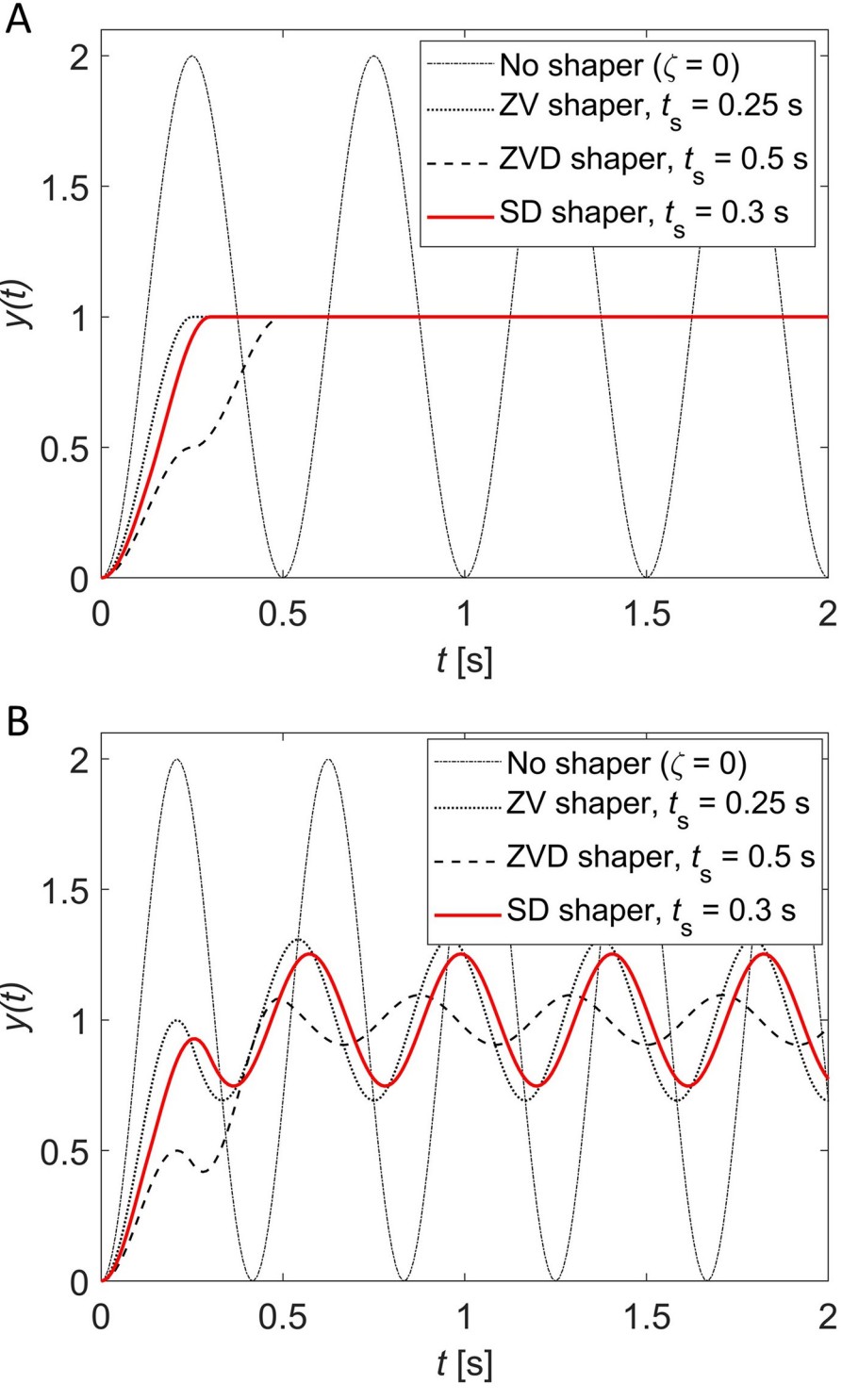

**Fig 5. Unit-step responses when the SD shaper with $t_s$ = 0.3s is used for an undamped system $\omega_n^2/(s^2 + \omega_n^2)$ with $\hat{\omega}_n$ = 2 Hz.** (A) No modeling error ($\omega_n = \hat{\omega}_n = 2$ Hz), (B) 20% modeling error in natural frequency ($\omega_n = 2.4$ Hz, $\hat{\omega}_n = 2$ Hz).

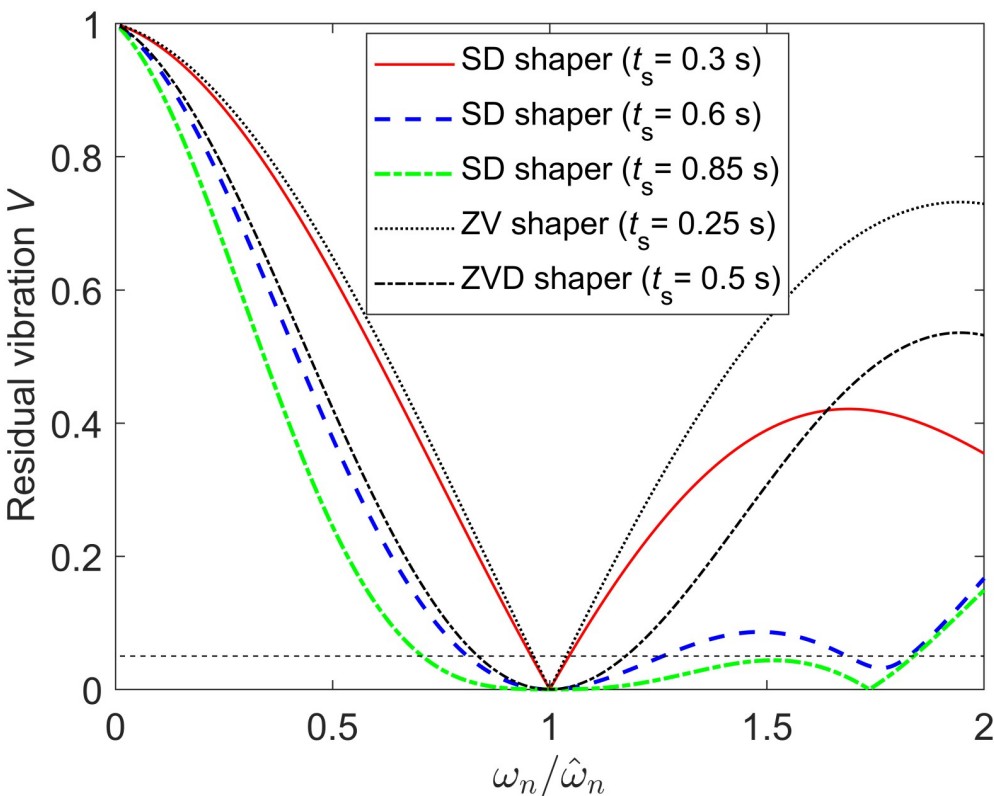

**Fig 6. Sensitivity curves of the SD shapers with shaping time $t_s$ = 0.3, 0.6, 0.85s for an underdamped system** $\omega_n^2 / (s^2 + 2\zeta\omega_n s + \omega_n^2)$ **with** $\hat{\omega}_n = 2$ **Hz and** $\zeta = 0.1$. Because the dimensionless shaping times are 0.597, 1.194, and 1.692, SD shapers are composed of three, four, and five impulses, respectively, and 5% insensitivities $I_{0.05}$ are 0.088, 0.452, and 1.133, respectively.

the sensitivity curves of the ZV and ZVD shapers for comparison. In case of a shaping time of 0.3 s, $I_{0.05}$ of the undamped system is 0.073, whereas $I_{0.05}$ of the underdamped system with $\zeta$ = 0.1 is increased to 0.088. The $I_{0.05}$ increased because the residual vibration $V$ decreased owing to increased damping.

Fig 7 shows unit-step responses when these SD shapers are applied to the underdamped system with $\zeta$ = 0.1. Fig 7A shows unit-step responses when no modeling error exists ($\omega_n = \hat{\omega}_n = 2$ Hz), and Fig 7B shows unit-step responses when 20% modeling error exists in natural frequency ($\omega_n = 2.4$ Hz, $\hat{\omega}_n = 2$ Hz). If no modeling error exists, the SD shapers exhibit the exact shaping time of 0.3, 0.6 and 0.85 s as designed in Eqs (23), (25) and (27), respectively. If 20% modeling error occurs in natural frequency, the magnitude of residual vibration decreases as the shaping time increases as shown in Fig 7B, which reveals the same trend as the sensitivity curve in Fig 6.

If $t_s$ = 0.85 s for the underdamped system with $\hat{\omega}_n = 2$ Hz and $\zeta$ = 0.1, the dimensionless shaping time is $T_s \approx 1.692 > 1.5$, and thus, the SD shaper is composed of five impulses according to (7). However, considering the impulse vector diagram for the case with $T_s > 1.5$, an input shaper with three, four or six impulses can also be designed because the sum of the corresponding impulse vectors can be made to be zero. Table 1 shows the SD shaper with five impulses and three input shapers with three, four and six impulses obtained using design procedures (ii) and (iii) and excluding procedure (i) in the previous section. However, this input shaper with three, four or six impulses is not a satisfactory solution because of its high

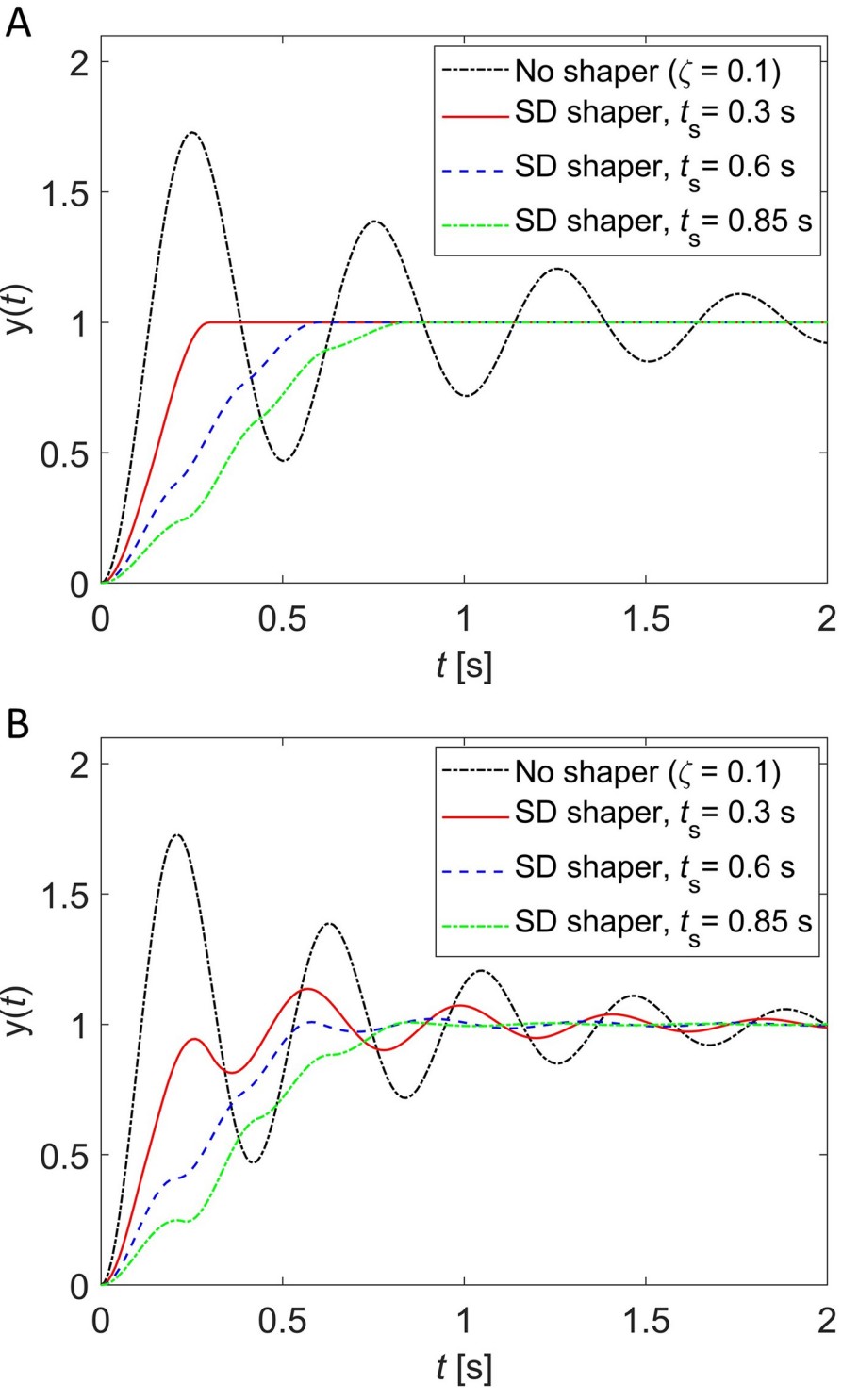

**Fig 7. Unit-step responses when SD shapers with shaping time $t_s$ = 0.3, 0.6, 0.85s are used for an underdamped system $\omega_n^2/(s^2 + 2\zeta\omega_n s + \omega_n^2)$ with $\hat{\omega}_n = 2$ Hz and $\zeta = 0.1$.** (A) No modeling error ($\omega_n = \hat{\omega}_n = 2$ Hz), (B) 20% modeling error in natural frequency ($\omega_n = 2.4$ Hz, $\hat{\omega}_n = 2$ Hz).

**Table 1. The SD shaper with five impulses and input shapers with three, four and six impulses for the shaping time $t_s$ = 0.85s.**

| Input shapers | Shaping time $t_s$ = 0.85s |
|---|---|
| SD shaper with five impulses | $\begin{bmatrix} t_i \\ A_i \end{bmatrix} = \begin{bmatrix} 0, & 0.2273, & 0.4346 & 0.6335, & 0.85 \\ 0.1436, & 0.3301, & 0.3060, & 0.1756, & 0.0447 \end{bmatrix}$ |
| Input shaper with four impulses | $\begin{bmatrix} t_i \\ A_i \end{bmatrix} = \begin{bmatrix} 0, & 0.2838, & 0.5715, & 0.85 \\ 0.2198, & 0.3993, & 0.2950, & 0.0859 \end{bmatrix}$ |
| Input shaper with three impulses | $\begin{bmatrix} t_i \\ A_i \end{bmatrix} = \begin{bmatrix} 0, & 0.2473, & 0.85 \\ 0.5754, & 0.4143, & 0.0103 \end{bmatrix}$ |
| Input shaper with six impulses | $\begin{bmatrix} t_i \\ A_i \end{bmatrix} = \begin{bmatrix} 0, & 0.2182, & 0.3514, & 0.3532, & 0.6001, & 0.85 \\ 0.0547, & 0.0534, & 0.0274, & 0.3272, & 0.4160, & 0.1237 \end{bmatrix}$ |

sensitivity to modeling errors compared to the SD shaper. Fig 8 shows that the SD shaper with five impulses for $t_s$ = 0.85 s is less sensitive to modeling errors than the input shaper with three, four or six impulses for $t_s$ = 0.85 s since the $V$ values of the SD shaper are smaller than the $V$ values of the three input shapers around $\omega_n / \hat{\omega}_n = 1$ in Fig 8. The main reason for the improved robustness of the SD shapers than input shapers with less impulses for the same shaping time is that SD shapers satisfy more derivative constraints than input shapers with less impulses. However, it may not be possible to obtain an input shaper with six impulses because all nine constraints [Eq (26) plus two more third-order derivative constraints] cannot be

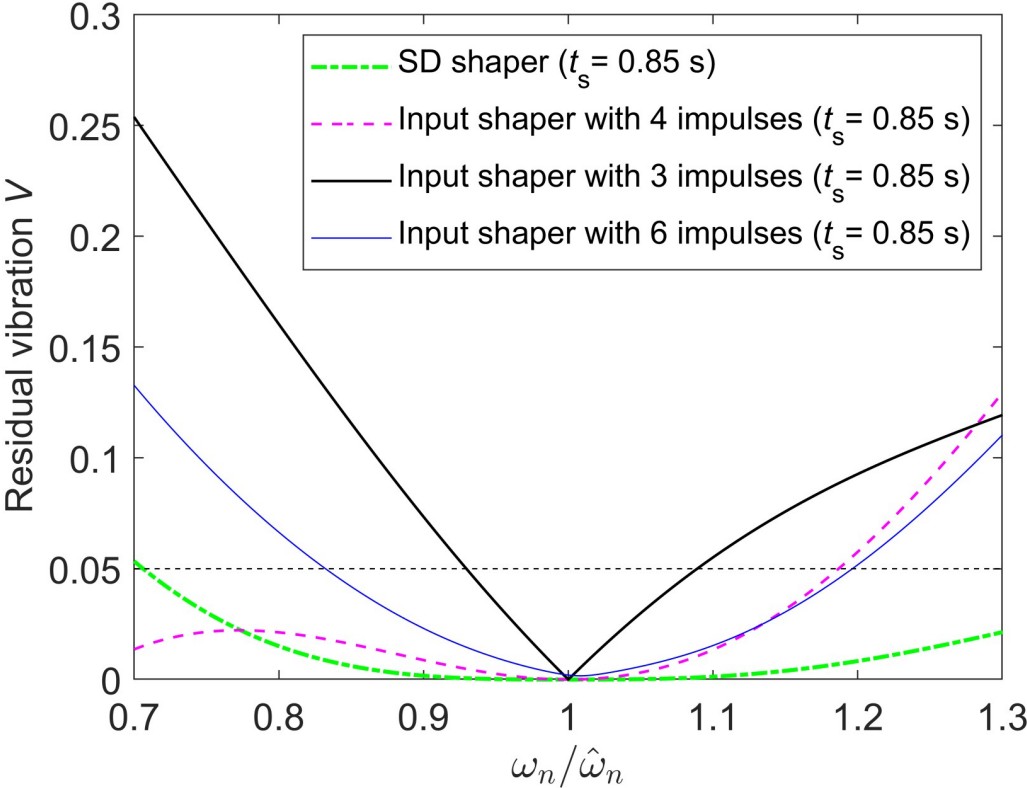

**Fig 8. Sensitivity curves of the SD shaper with five impulses and input shapers with three, four or six impulses for the case with the shaping time $t_s$ = 0.85 s for an underdamped system with $\hat{\omega}_n = 2$ Hz and $\zeta = 0.1$.** The SD shaper with five impulses is more robust to modeling errors than the input shaper with three, four or six impulses for $t_s$ = 0.85 s.

satisfied for the case with $t_s$ = 0.85 s. Even if we find an input shaper with six impulses for $t_s$ = 0.85 s numerically, it may be an approximate solution with less robustness such that the sum of $A_i$ is close to one (not exact one) as can be checked in the last input shaper of Table 1. Note that the sum of $A_i$ of the first three input shapers in Table 1 are exactly one, but the sum of $A_i$ of the last input shaper is 1.0024.

## Experimental demonstrations

A horizontal beam vibration apparatus that can control a flexible horizontal cantilever beam vertically was used to verify the validity of the SD shapers (Fig 9) [28]. In the apparatus, vertical motions of the flexible beam were generated by controlling a servo motor with a timing belt. A laser sensor was placed under the tip of the beam to collect motion data. Note that the sensor data were not used to control the flexible beam because the input shaper is the open-loop control method. To enhance the effect of the residual vibration, masses were attached at the end of the flexible beam.

The block diagram of the horizontal beam vibration apparatus is shown in Fig 10. As shown in Fig 10, the major three components of the block diagram are a proportional-and-derivative (PD) controller, motor dynamics, and beam dynamics. The PD feedback controller $u(t) = K_p[e(t) + T_d\dot{e}(t)]$ with $K_p$ = 0.18 and $T_d$ = 0.075 was used to control the vertical motion of the moving block. The motor dynamics was modeled as a first-order system with a time constant of 16 ms. Here, $K_{LG}$ represents the conversion factor of the linear guide with the timing belt, which generates linear motion from the motor rotation. The flexible horizontal beam dynamics was modeled as a second-order underdamped system $\omega_n^2 / (s^2 + 2\zeta\omega_n s + \omega_n^2)$.

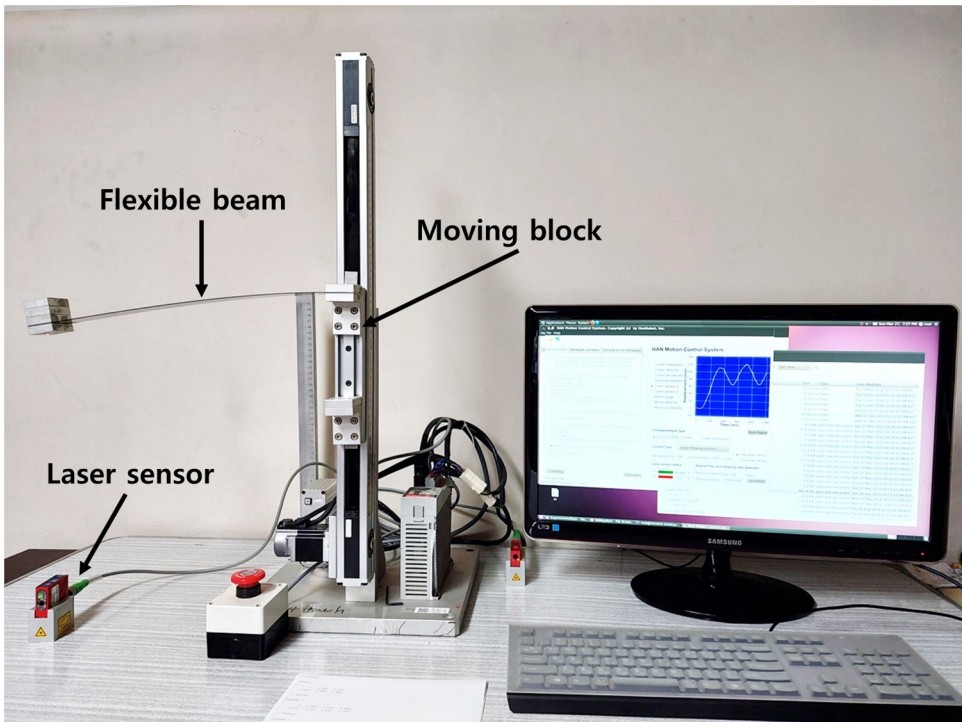

**Fig 9. Horizontal beam vibration apparatus.** The apparatus can generate vertical movements with a flexible horizontal cantilever beam with added masses at the end. The system has natural frequency $\omega_n$ of 16.7 Hz and damping ratio $\zeta$ of 0.002.

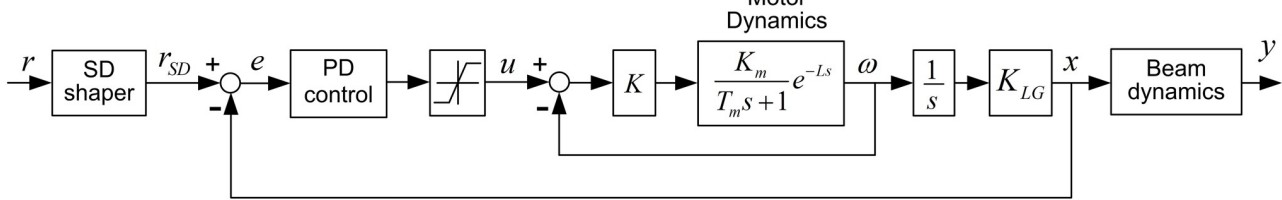

**Fig 10. Block diagram of the horizontal beam vibration apparatus.** The system includes a PD controller, motor dynamics, and beam dynamics.

To validate the design methodology and performance of the SD shaper, three SD shapers with three shaping times, namely 0.2, 0.5, and 0.7 s, were used. To obtain SD shapers, determining the natural frequency $\omega_n$ and damping ratio $\zeta$ of the system, which is the horizontal beam vibration apparatus in the experiments, is crucial. Based on 10 s of the step response of the flexible beam with no input shaper, the natural frequency and damping ratio were modeled to be 16.7 Hz and 0.002, respectively. With these modeled values, the damped period $T_d$ was calculated to be 0.3762. The black dotted line in Fig 11 represents the experimental step response of the flexible beam with no input shaper.

Based on Eq (6), the SD shaper with a shaping time of 0.2 s has the dimensionless shaping time $T_s$ of 0.53, and then, according to Eq (7), three impulses are required ($N = 3$). In the same manner, the SD shaper with a shaping time of 0.5 s requires four impulses ($N = 4$) because $T_s$ is 1.33, and the SD shaper with a shaping time of 0.7 s requires five impulses ($N = 5$) because $T_s$ is 1.86. Based on these parameters, the SD shaper for each case was obtained by solving numerically using MATLAB. Table 2 summarizes the obtained three SD shapers.

To validate the performance of the obtained SD shapers, both simulations and experiments were performed with a 100 mm step input displacement, and step responses were compared. The three obtained SD shapers were simulated using Simulink, and the results are shown in Fig 11. The simulations were performed according to the block diagram displayed in Fig 10. The solid blue, red, and green lines in Fig 11 depict the simulation results for SD shapers with shaping times of 0.2, 0.5, and 0.7 s, respectively. The SD shapers were experimented using the apparatus, and the results are plotted in Fig 11 for comparison. The dotted blue, red, and green lines in Fig 11 denote the experimental results of the SD shapers with 0.2, 0.5, and 0.7 s shaping times, respectively. As the specified shaping time is increased, the actual shaping time is also increased in both simulation and experimental results, as expected. In Fig 11, the experimental results of the SD shapers have the same trend as the simulation results with very little discrepancy.

Fig 12 shows the simulation results to compare the effect of the motor feedback dynamics for the obtained SD shapers. For both solid and dashed lines in Fig 12, red, blue, and green colors represent the results of SD shapers with shaping times 0.2, 0.5, and 0.7 s, respectively. The dashed lines represent the results of SD shapers when the system model does not include motor feedback dynamics, and thus, the system model only consists of the SD shaper and beam dynamics. The solid lines represent the results of SD shapers when the system model includes motor feedback dynamics, which is the system model based on the experimental apparatus. The solid lines in Fig 12 are the same as the simulation results of SD shapers in Fig 11. The dashed lines in Fig 12 reveal that the SD shapers can suppress residual vibration in a planned specified shaping time. However, the solid lines in Fig 12 show that the shaping times of the SD shapers are slightly longer than the specified shaping times because of the motor feedback dynamics. The experimental and simulation results reveal that SD shapers with arbitrarily specified shaping times can eliminate the residual vibrations of a flexible beam as expected.

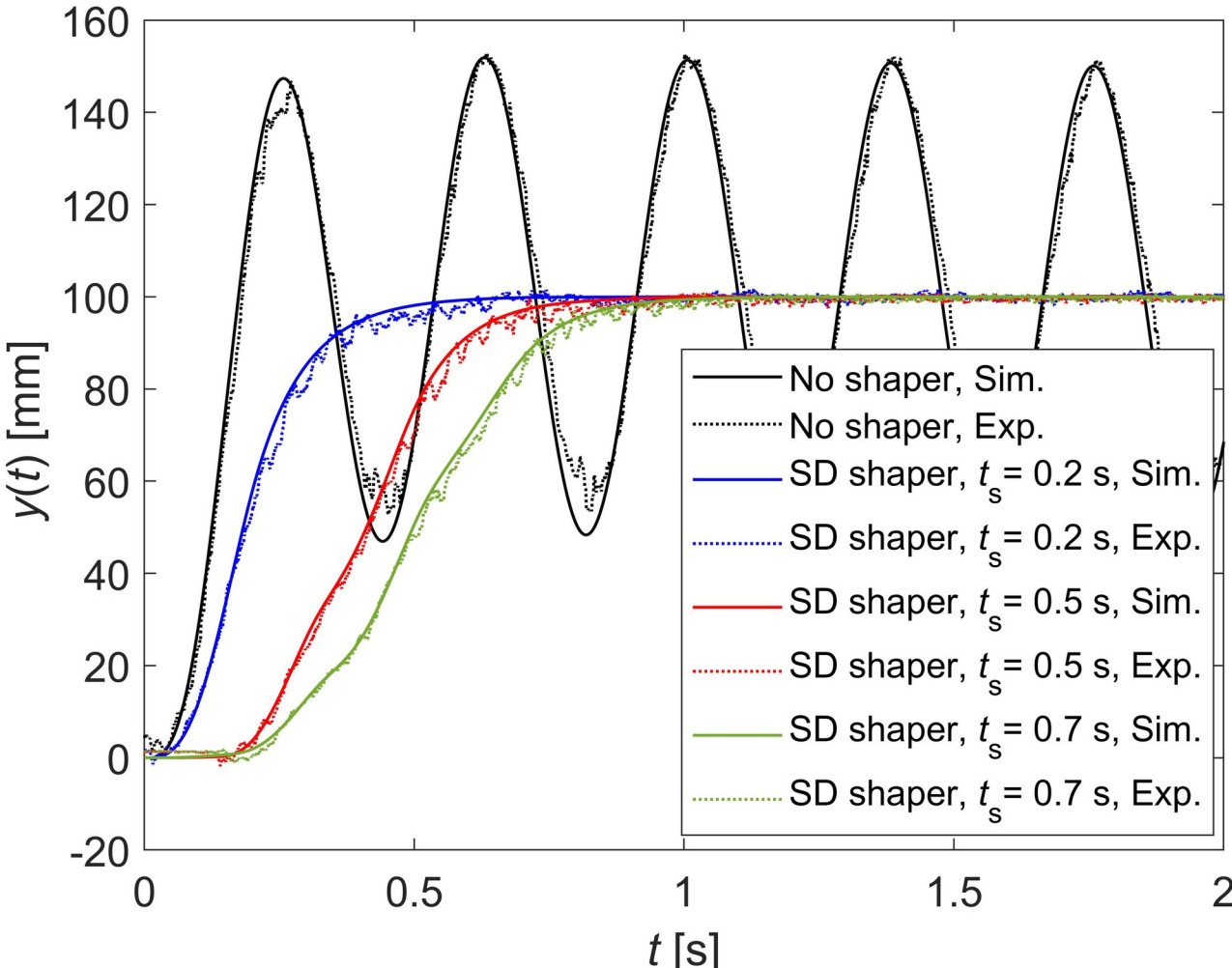

**Fig 11. Step responses through simulations and experiments.** The command input of step responses is 100 mm step. Black: No input shaper. Blue: SD shaper with 0.2 s shaping time. Red: SD shaper with 0.5 s shaping time. Brown: SD shaper with 0.7 s shaping time.

## Conclusion

This paper introduced an SD shaper that can suppress residual vibrations at an arbitrarily specified shaping time with the largest 5% insensitivity to modeling errors when the

**Table 2. Obtained SD shapers with three shaping times.**

| Shaping time | N | SD shaper |
|---|---|---|
| 0.2 s | 3 | $\begin{bmatrix} t_i \\ A_i \end{bmatrix} = \begin{bmatrix} 0, & 0.0610, & 0.2 \\ 0.4123, & 0.1109, & 0.4768 \end{bmatrix}$ |
| 0.5 s | 4 | $\begin{bmatrix} t_i \\ A_i \end{bmatrix} = \begin{bmatrix} 0, & 0.1251, & 0.3122, & 0.5 \\ 0.0039, & 0.2532, & 0.4971, & 0.2459 \end{bmatrix}$ |
| 0.7 s | 5 | $\begin{bmatrix} t_i \\ A_i \end{bmatrix} = \begin{bmatrix} 0, & 0.1430, & 0.3261 & 0.5126, & 0.7 \\ 0.0089, & 0.1387, & 0.3750, & 0.3601, & 0.1172 \end{bmatrix}$ |

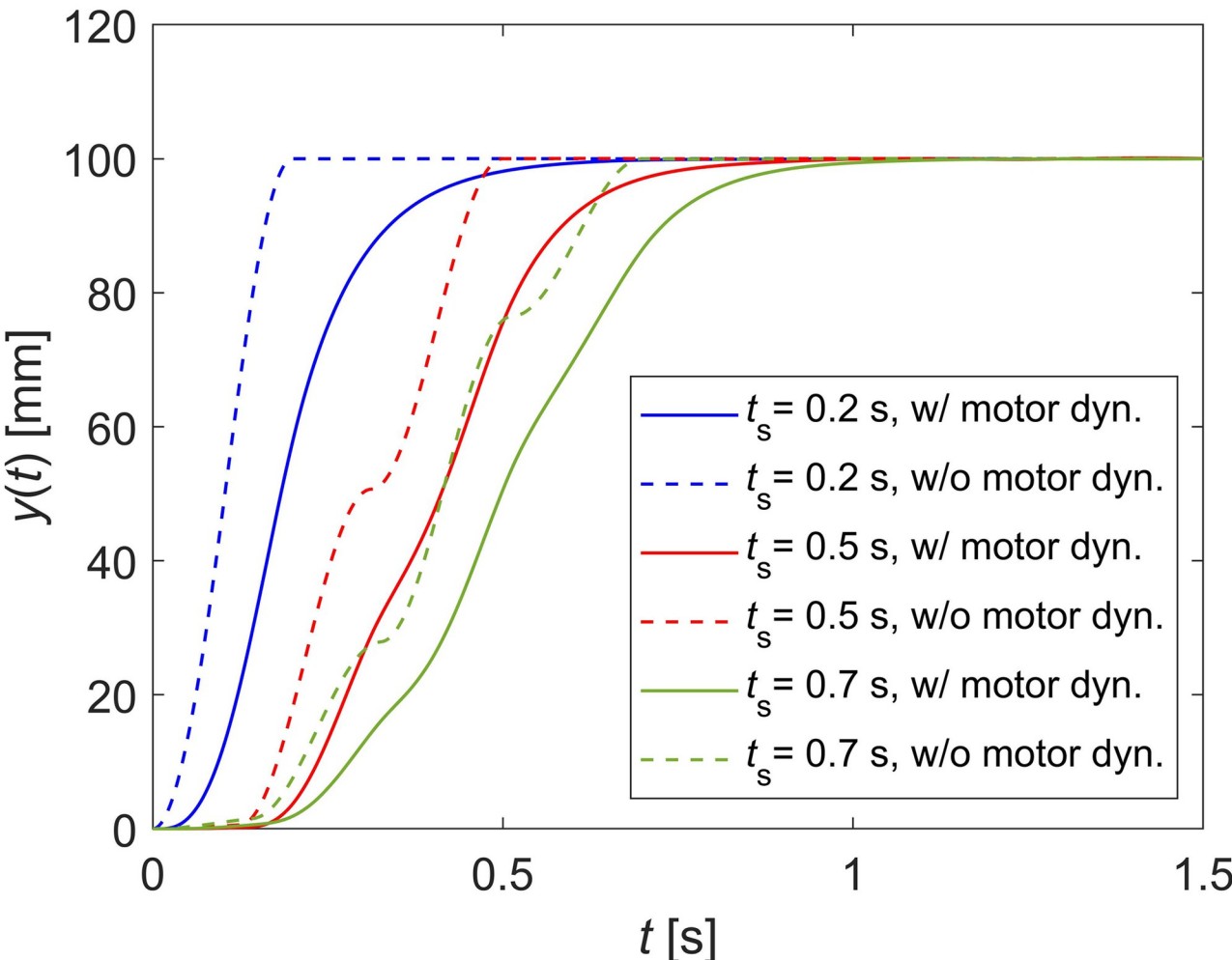

**Fig 12. Simulation results for three SD shapers with and without motor feedback dynamics.**

magnitude of the last impulse is taken as a free parameter. From the specified shaping time, the angle of the last impulse vector was calculated, and the number of impulses was determined from the dimensionless shaping time. Then, by solving the constraint equations of the SD shaper analytically or numerically, the desired SD shaper was obtained. As the specified shaping time increases, the SD shaper increases the number of impulses one by one according to the number of added derivative constraints to improve the robustness to modeling errors.

The performance of the SD shapers for specified shaping times were evaluated through simulations by plotting sensitivity curves for modeling errors and step responses of the second-order system. The simulations revealed that the SD shapers suppress residual vibrations of the vibratory system at the specified shaping times, and the robustness of the SD shapers to modeling errors improves as the shaping time increases. Furthermore, the validity of the SD shapers was experimentally verified using a horizontal beam vibration apparatus.

For further study, the SD shaper proposed in this paper could be extended to a negative SD shaper or a multi-mode SD shaper.

## Acknowledgments

The author acknowledges Prof. Manh-Tuan Ha and Prof. Chul-Goo Kang for lending the horizontal beam vibration apparatus for experimental works.

## Author Contributions

**Writing – original draft:** Brian Byunghyun Kang.

**Writing – review & editing:** Brian Byunghyun Kang.

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
