## [Decision Letter · Decision Letter 0]

9 Aug 2022

PONE-D-22-16643Specified-duration shapers for suppressing residual vibrationsPLOS ONE

Dear Dr. Kang,

Thank you for submitting your manuscript to PLOS ONE. After careful consideration, we feel that it has merit but does not fully meet PLOS ONE’s publication criteria as it currently stands. Therefore, we invite you to submit a revised version of the manuscript that addresses the points raised during the review process.

We look forward to receiving your revised manuscript.

Kind regards,

Ning Cai, Ph.D.

Academic Editor

PLOS ONE

Journal Requirements:

    "This work was supported by the National Research Foundation of Korea (NRF) grant funded by the Korea government (MSIT) (No. 2021R1G1A109521912). "

Reviewers' comments:

Reviewer's Responses to Questions

**Comments to the Author**

1. Is the manuscript technically sound, and do the data support the conclusions?

Reviewer #1: Yes

Reviewer #2: Yes

Reviewer #3: Yes

Reviewer #4: Yes

2. Has the statistical analysis been performed appropriately and rigorously? 

Reviewer #1: N/A

Reviewer #2: Yes

Reviewer #3: N/A

Reviewer #4: N/A

3. Have the authors made all data underlying the findings in their manuscript fully available?

Reviewer #1: Yes

Reviewer #2: Yes

Reviewer #3: Yes

Reviewer #4: Yes

4. Is the manuscript presented in an intelligible fashion and written in standard English?

Reviewer #1: Yes

Reviewer #2: Yes

Reviewer #3: Yes

Reviewer #4: No

5. Review Comments to the Author

Reviewer #1: I read your paper with interest and enjoyed it.

I have two major concerns with the paper. First of all, I do not think that you have given a clear description of the problem you are trying to solve. You state (in lines 70-73) that "In this paper, a novel specified-duration (SD) shaper that satisfies arbitrarily specified shaping time is proposed. The SD shaper is an input shaper that is designed to remove residual vibration during a given specified shaping time. A systematic method is presented to design the SD shaper with an arbitrarily specified shaping time by using impulse vectors." You do not state that you impose any constraints on the sensitivity to changes in the parameters that describe the system. In the body of the paper, as you increase the "dimensionless shaping time," you also increase the number of impulses used and you add constraints to force the sensitivity to be small.

It seems clear that you want the SD shaper to be insensitive to parameter mismatches, but you do not make that a stated goal. This goal needs to be stated clearly at the very beginning of the paper.

This additional, but almost unmentioned, constraint leads to other problems. For example, if one wants the duration of an input shaper to be some number of half-periods of the system’s response, you can design an input shaper with two impulses. This is not mentioned -- and, in fact, in equation (6), you seem to say otherwise. In line 136, you state that you do not design input shaper with N=3 when Ts > 1 even though you could because they are quite sensitive to modeling error, but this was never an "official" constraint.

My second major concern has to do with the “design rules” for the SD shaper. If I understood what you did correctly, you designed a series of input shapers by specifying a duration and then choosing a number of impulses and relevant constraints that experience showed worked well. You made sure that you had one “free” coefficient, and iterated on it to find the value that led your SD shaper to be as insensitive to parameter mismatch as possible. If this is correct, you should describe the technique first, and you should explain how you chose the number of impulses – since fewer impulses could have been used.

Also, as things stand, the design rules seem very “ad hoc.” If you can provide a better explanation for where they come from and how you decided how many impulses to use, that would be most helpful to the reader and would strengthen the paper.

I would like to raise several somewhat more minor issues as well.

On line 32, you speak of two researchers as being at MIT. As far as I can tell, neither is at MIT today.

On line 54, I think that “conventionally” should be replaced with a more suitable word.

Equations (1) and (2) did not print properly in the PDF.

You seem to assume a second order system. As I think that this is actually part of what sets the number of impulses, this assumption should also be made clear quite early on. If you have also examined systems with more than one pair of complex poles, you can say a bit about such systems too.

On lines 93-96 you give a description of the Dirac delta function. I think this description can be omitted.

Is it well known that your non-linear design equations always have a solution? If it is, it would be nice to state the fact and give a reference to it.

On line 129 and in other places, you speak of having a “predetermined” value. It seems that what you mean is that this value can be chosen by the designer, and when you later step through values of a parameter, it is this parameter that you change. If this is correct, then I think that the word “predetermined” is somewhat misleading and should be changed.

Your conclusion also does not mention insensitivity to changes in the system’s parameters. As dealing with this issue is one of the design constraints, it should be mentioned in the conclusion as well.

Overall, I enjoyed the article, but its goals must be stated precisely and the design rules and their motivation must be explained clearly. I also think that the article could also be shortened somewhat. There are more examples of how to calculate SD shapers than seem necessary to me.

Reviewer #2: This paper is one application of impulse vectors for input-shaping control. The finding is predictable, but may be useful for some applications that require limited response time, and of course, the vibration suppression performance is limited. In comparison to traditional IS techniques such as ZV, ZVD, more computation is required. But it helps readers to understand the input shaping technique more clearly. It also shows the effectiveness of impulse vector approach to generalize the development of input shaping technique.

Reviewer #3: The paper aims to define a particular class of Input shaping filters with a specified-duration (SD). Instead, classical input shapers only take into account the characteristics of the system to which they are applied. So the total duration of these shapers is not a free parameter for the designer. First, the procedure to define the SD shapers is defined, then these filters are analyzed with both a numerical and an experimental approach. The main contributions of this work consist in the definition of the SD shapers.

I will focus only on the contents of the paper and not on its form. The article is interesting, but I think that some improvements are necessary before publication.

1) Lines 39/40: “The vibratory system in Fig. 1 includes a feedback loop and a feedback controller”. I think it could be useful to show the feedback loop also in the figure, since talking about it without showing it can confuse the reader.

2) Lines 55/57: “The SI shaper was designed to ensure that the sensitivity function V of the input shaper was within 0.05 for a specified range of modelling errors”. The value 0.05 (i.e. 5%) is only exemplary. An SI shaper can be modeled with other percentages of admitted sensitivity. I think it is better to use in all the text (not only in these lines) a generic value (e.g. called z) instead of 0.05 and to specify the entity of this value only when it is strictly necessary: e.g. in the numerical examples to calculate the parameters of the shapers it is necessary to use z = 0.05 (or z equal to another numeric value).

3) Lines 67/69: “Although the shaping time is an important factor in the design of input shapers, no studies have been conducted on designing input shapers that can satisfy arbitrarily specified shaping times”. In [1,2] classical input shapers (ZV, ZVD, …) are used to obtain motion laws with a prescribed duration. The procedure adopted in this work is very different since here the shaper is directly designed with a prescribed duration. In [1,2], instead, the shapers are computed, and then, by knowing their duration, the motion law in input to the shaper is found in such a way to execute the movement in the right total time. Even if the method is different, I think that the final aim of the input shapers defined in this manuscript could be the same obtained in [1,2]. With this in mind, I think an analysis of the differences between the method adopted in this work and the technique applied in [1,2] could be interesting. Is the solution proposed in this paper better/worse/indifferent?

[1] Boscariol, P., and Richiedei, D., 2018, “Robust Point-to-Point Trajectory Planning for Non-Linear Underactuated Systems: Theory and Experimental Assessment,” Rob. Comput. Integr. Manuf., 50, pp. 256–265.

[2] Guagliumi, L., Berti, A., Monti, E., and Carricato, M.: Anti-Sloshing Trajectories for HighAcceleration Motions in Automatic Machines. ASME, Journal of Dynamic Systems, Measurement, and Control (2022). doi: 10.1115/1.4054224.

4) Line 92. I suggest writing relation ω_d=ω_n √(1-ζ^2 ) as an equation and not writing it in the text.

5) Lines 120/134: “If the dimensionless shaping time Ts is between 0.5 and 1 (that is, between half and a period), the SD shaper consists of three impulses. If Ts is between 1 and 1.5, the SD shaper consists of four impulses, and if it is between 1.5 and 2, it consists of five impulses. …” Why? It is not clear how the number of impulses is chosen, but this is an important point in the definition of the SD shaper. It is necessary to better explain the reason behind the choice of the number of impulses.

6) Lines 158/159: “Among them, A1, A2, A3, t2 with the biggest 5% insensitivity I0.05 is selected as the SD shaper.” I think this sentence confuses the reader since the explanation of how these values are chosen to maximize the insensitivity of the shaper is given in the following. I suggest omitting this sentence.

7) Lines 197/203. Is it possible to find an analytical solution to the problem of finding the shaper that maximizes the insensitivity? Maybe by adding another constraint to the original ones in equations (7)-(9) to find the value of A3. The resulting system could be solved also numerically. If this is not possible, this should be discussed: why is it not possible?

8) Line 252. In equation (23), constraints 3 and 4 should be clarified. It is mentioned that it is necessary to add derivative constraints, but more specifically, how these constraints are computed? I think they are the derivative of constraints 1 and 2 with respect to ω_n, but this should be clarified. Idem in Eq (25).

9) Lines 299-300: “Moreover, if the dimensionless shaping time is less than 0.5, we require negative impulses to remove residual vibrations.” Why? It would be interesting to explain the reason behind this consideration.

10) Lines 324/325: “If no modeling error exists, then the SD shaper has the shaping time of exactly 0.3 s as shown in Fig 5A”. I thought the duration of the shaper was a free parameter chosen by the designer. If there are errors in the modeling of ω_n, this should not interfere with the duration of the shaper. Am I right? If yes, I would omit this sentence that confuses the reader. Idem for lines 354/355.

11) Line 409/410: “The obtained three SD shapers were simulated using Simulink, and the results are shown in Fig 10”. I think it would be interesting to show also in the experimental results a comparison between the SD shaper and the classical ZV/ZVD as previously done for the numerical results related to the sensitivity analysis.

Some improvements are necessary before the publication. In particular, I think it is very important to be more clear on point 5) in my list since the choice of the number of impulses of the input shaper is one of the most important factors in the definition of the shaper, but in my opinion here this aspect is not clear.

Reviewer #4: The paper suggest a new input shaper with arbitrarily specified shaping times, a specified-duration shaper. The author test their proposed shaper using numerical simulation of a second-order system and validate it experimentally by suppressing the induced residual vibration of a horizontal beam. The paper is not recommended to be published in this form for mainly two reasons; (1) the novelty of the proposed shaper where several shaped commands using optimal control and command shaping were recently developed with robustness adjustability, smoothness enhancement, and variable command time length, and (2) the presentation of the paper including its writing style and structure. For instance, the author can consult the following papers for more details:

“Robust time-delay control of multimode systems” by Singh and Vadali.

“Discrete-time command profile for simultaneous travel and hoist maneuvers of overhead cranes” by Alghanim et. al.

Here are my comments to the authors:

The readers of published papers are aware of the vibration types so no need to mention that there free and force vibration in page 2 line 24.

There is no need to mention that Smith was at UC Barkeley when he introduced the posicast control; page 2 line 32.

The same previous comment goes with Singer and Singhose at MIT.

Page 2 line 33; I am not sure the readers will know what the derivative constraints is. The author could simply state the Singer and Singhose enhance the robustness.

Omit the sentence “in natural frequency and damping ratio” in page 2 line 37.

The last sentence in the second paragraph is redundant.

The open-loop control is also sensitive to external disturbance.

The last sentence in the third paragraph is redundant.

The first and second sentences is page 3 (lines 47 and 48) are redundant.

The paragraph starting from the line 55 in page is confusing since the readers may not know what the sensitivity function V is.

The introduction is quite weak and need to be improved. The author should survey through recent papers in command and input shapings to further demonstrate the novelty of the present results.

The equations (1) and (2) are only limited to underdamped system.

The sentences starting from line 93 should be omitted because it is quite obvious what impulses are and what dirac delta function is.

The equations (4) and (5) are obvious and should be omitted.

The paragraph starting from line 127 should be rephrased.

No need to mention what kind of software the author used since the iterative problem is quite easy and straightforward.

The something goes when the author stated that they used the MATLAB built-in function fsolve to solve the nonlinear problems. I am quite sure that the resultant system of nonlinear equations are not really complex and so need to explicitly state what kind of solver the author used to solve for the roots.

The SP shapers were only designed when 0.5<t_s<1 1="" and=""> Write the command duration of the ZV and ZVD in all the figures to have reasonable comparison.

Provide a thorough numerical experiment and summarize the findings at the end of the numerical and experiment sections.

The paper can be further improved if the author attach the general case where special cases can be derived from the general case. The author should also read and survey the recent contributions of controlling techniques especially the ones dealing with the input and command shaping. The language and the presentation of the paper should improved. Obvious phrases, simple equations, and repeated sentences should be omitted in order to make the reader interesting when he/she is reading the paper. To make it simple, the proposed shaper could be published but the author should try to convince the reviewers the novelty of his shaper and the contribution of his findings.</t_s<1>

6. PLOS authors have the option to publish the peer review history of their article (what does this mean?). If published, this will include your full peer review and any attached files.

Reviewer #1: No

Reviewer #2: No

Reviewer #3: No

Reviewer #4: No

---

## [Author Response · Author response to Decision Letter 0]

23 Aug 2022

Since the response letter includes equations and pictures, "Response to Reviewers" file is uploaded as a seperate file as requested.

---

## [Decision Letter · Decision Letter 1]

12 Sep 2022

PONE-D-22-16643R1Specified-duration shapers for suppressing residual vibrationsPLOS ONE

Dear Dr. Kang,

Thank you for submitting your manuscript to PLOS ONE. After careful consideration, we feel that it has merit but does not fully meet PLOS ONE’s publication criteria as it currently stands. Therefore, we invite you to submit a revised version of the manuscript that addresses the points raised during the review process.

We look forward to receiving your revised manuscript.

Kind regards,

Ning Cai, Ph.D.

Academic Editor

PLOS ONE

Additional Editor Comments:

I attach the review reports. If some reviewer recommends a lot of publications, you can decide by yourself whether they are really relevant. There's no need to do everything in the comments.

Reviewers' comments:

Reviewer's Responses to Questions

**Comments to the Author**

1. If the authors have adequately addressed your comments raised in a previous round of review and you feel that this manuscript is now acceptable for publication, you may indicate that here to bypass the “Comments to the Author” section, enter your conflict of interest statement in the “Confidential to Editor” section, and submit your "Accept" recommendation.

Reviewer #1: (No Response)

Reviewer #3: All comments have been addressed

Reviewer #4: (No Response)

2. Is the manuscript technically sound, and do the data support the conclusions?

Reviewer #1: Partly

Reviewer #3: Yes

Reviewer #4: Yes

3. Has the statistical analysis been performed appropriately and rigorously? 

Reviewer #1: N/A

Reviewer #3: N/A

Reviewer #4: N/A

4. Have the authors made all data underlying the findings in their manuscript fully available?

Reviewer #1: Yes

Reviewer #3: Yes

Reviewer #4: Yes

5. Is the manuscript presented in an intelligible fashion and written in standard English?

Reviewer #1: Yes

Reviewer #3: Yes

Reviewer #4: Yes

6. Review Comments to the Author

Reviewer #1: Referee’s Report

Title: “Specified-duration shapers for suppressing residual vibrations”

Author: B. B. Kang

I read your paper with interest. I still feel that the point of the paper is not being presented well. You are presenting a method to design a controller that is both specified duration and insensitive to changes, yet you call your controller an SD controller and do not stress that it is also required to be relatively insensitive. Because you do not stress that the system must be insensitive, many of the choices you make are hard to understand.

One place you could stress the “robustness” issue is on line 15. There you refer to the shaping time. I think it would be best to refer to the shaping time and to the robustness.

I would strongly recommend changing the name of the controller. SD puts all the emphasis on the duration and none on the “robustness.” I think that if you change the emphasis to include robustness, you will help the reader understand what you are doing.

From roughly line 140 to line 155, you describe some of your design rules. On lines 141 and 144 you state that four or five (respectively) impulses are, _therefore_ [my emphasis], required. On lines 150 – 153, you state that you can actually manage with three impulses. Thus, four or five impulses are not required, and the word “therefore” is misleading.

On lines 152 and 153, you state that the three impulse solution is unacceptable because it is not sufficiently insensitive. Once again, we see that it is critical that you state that low-sensitivity is an absolute requirement from your input shaper. Then you must explain or demonstrate that the three impulse solution is too sensitive to modeling errors.

If I understand what you are trying to do correctly, you want an insensitive specified duration shaper. It seems that experience/experiments have shown how many impulses are necessary in order to have enough degrees of freedom to achieve all your goals. If this is the case, please say so and provide theoretical or experimental support your statement. Currently, it is unclear how you arrived at (6).

As I mentioned in my previous report, the procedure here seems somewhat “ad hoc,” which is not necessarily a bad thing. The idea of adding a time constraint and then trying to produce a reasonably “insensitive” input shaper by using several impulses and impose constraints on the input shaper is interesting. If the results are largely experimental, say so and provide tables and figures to compare your results to results attained using more standard methods. Show the reader that your design methods lead to practical input shapers that outperform the well known ones.

Minor Issues

Why do you not try to add impulses even for smaller values of T_s? Would that not allow you to make the input shaper more “robust?”

On line 17, the word “are” should be replace by “is”.

On line 27, I would say that one often wants to reduce vibrations. Sometimes, one is actually trying to build an oscillator.

On line 35, you should say “Since the 1980s”. (You need to add the word “the.”) Also, please note that in your current bibliography, you have additional articles by some of the authors mentioned.

On line 42, I would say “an input shaper is a linear time-invariant system how impulse response consists of…”.

On line 80, I would use “performance” rather than “validity.”

On line 342, please replace “if actual” with “if the actual”.

Please copyedit the article carefully before resubmitting. (This is almost always helpful. )

Reviewer #3: The results described in the manuscript are predictable, but the paper could be useful for a reader interested in input shaping methods. Moreover, it is easy to understand and well written. The author has adequately addressed my comments, and I think this has been useful to improve the quality of the paper, which is now adequate for publication.

Reviewer #4: The author addresses several of my comments and fail to show the novelty of the proposed shaper especially surveying the literature. I will assure the paper could be published if the author either extend his shaper to multi-mode system or even explain how can be extended. One way is by using convolution where the other can be based on solving a system of nonlinear equations. Hence, the author could address this issue by reviewing the following papers:

1. “An extension of command shaping methods for controlling residual vibration using frequency sampling” by N. C. Singer and W. P. Seering.

2. “Using input command pre-shaping to suppress multiple mode vibration” by J. M. Hyde and W. P. Seering.

3. “Design of input shapers using modal cost for multi-mode systems” by R. Kumar and T. Singh.

4. “Command Shaping for Sloshing Suppression of a Suspended Liquid Container” by A. Alshaya and K. Alghanim

5. “Robust multi-steps input command for liquid sloshing control” by A. Alshaya and A. Alshayji.

6. “Robust time-delay control of multimode systems” by T. Singh and S. R. Vadali.

7. “Shaping Container Motion for Multimode and Robust Slosh Suppression” by B. Pridgen, K. Bai, and W. Singhose.

The advantages of the proposed SD shaper is its capability to be convolved with other filtering techniques; notch, low pass, or even band-gap filters. The author can for example consult the papers for Jie Huang who extend the input shaper to make it smooth and robust. Please see the following:

1. “Command Generation for Flexible Systems by Input Shaping and Command Smoothing” by W. Singhose, R. Eloundou, J. Lawrence.

2. “Vibration reduction for flexible systems by command smoothing” by X. Xie, J. Huang, and Z. Liang.

It is quite important that the author cites the new relevant papers to promote his work and demonstrate its innovative and not outdated.

Again, the author can either discuss in general how a negative impulse can added to his proposed SD shaper and how the SD shaper can be extended to multi-mode system. A paragraph stating the idea is fine. However, I, personally, recommend a demonstrated example especially the author states that he left where the experimental apparatus is, and he (the author) has no access now. Discussing these issues and showing the further capability of the proposed shaper (extension to multi-mode) will make the paper to be above the others. It is worth mentioning that not only the objective of a research is just to publish it but also to make it readable and citable.

General Comments:

1. Add the overmentioned papers in the introduction starting from the second sentence “for example the forced lateral vibration … “.

2. Add the paper entitled “An extension of command shaping methods for controlling residual vibration using frequency sampling” with references 17-19 since that paper presents EI shaper.

3. Even though I disagree with the need of writing the damped natural frequency as a new equation, but if the author prefers doing that please add an equation number.

4. I do not nee the author to re-produce the experimental results again. I only need the author to add the ZV and ZVD duration in the legends.

5. The same thing goes with the sensitivity curves.

7. PLOS authors have the option to publish the peer review history of their article (what does this mean?). If published, this will include your full peer review and any attached files.

Reviewer #1: No

Reviewer #3: No

Reviewer #4: No

---

## [Author Response · Author response to Decision Letter 1]

20 Sep 2022

Response letter to reviewers are uploaded as a seperate file. Thank you.

---

## [Decision Letter · Decision Letter 2]

26 Sep 2022

PONE-D-22-16643R2Specified-duration shapers for suppressing residual vibrationsPLOS ONE

Dear Dr. Kang,

Thank you for submitting your manuscript to PLOS ONE. After careful consideration, we feel that it has merit but does not fully meet PLOS ONE’s publication criteria as it currently stands. Therefore, we invite you to submit a revised version of the manuscript that addresses the points raised during the review process.

We look forward to receiving your revised manuscript.

Kind regards,

Ning Cai, Ph.D.

Academic Editor

PLOS ONE

Reviewers' comments:

Reviewer's Responses to Questions

**Comments to the Author**

1. If the authors have adequately addressed your comments raised in a previous round of review and you feel that this manuscript is now acceptable for publication, you may indicate that here to bypass the “Comments to the Author” section, enter your conflict of interest statement in the “Confidential to Editor” section, and submit your "Accept" recommendation.

Reviewer #1: (No Response)

Reviewer #3: All comments have been addressed

2. Is the manuscript technically sound, and do the data support the conclusions?

Reviewer #1: Partly

Reviewer #3: Yes

3. Has the statistical analysis been performed appropriately and rigorously? 

Reviewer #1: N/A

Reviewer #3: N/A

4. Have the authors made all data underlying the findings in their manuscript fully available?

Reviewer #1: Yes

Reviewer #3: Yes

5. Is the manuscript presented in an intelligible fashion and written in standard English?

Reviewer #1: Yes

Reviewer #3: Yes

6. Review Comments to the Author

Reviewer #1: I read the revised version of your paper with interest. I have several comments and suggestions. Number 7, below, is, in my opinion, very important as it provides a partial justification of your equation 7 -- which is central to your design procedure.

1. In several places (such as lines 16 and 75), you speak of “maximal robustness.” I think it would be better to speak of robustness without mentioning maximality. It is not clear that your controller is maximally robust.

2. In line 37, the citation [15-20] contains a paper not authored by Singer, Singhose, Singh, and it is missing some papers written by one or more of them that appear in your bibliography.

3. Before line 128, I think that you should explain that you need some flexibility in order to achieve robustness. In order to give yourself the flexibility you need, you add impulses. Then you can discuss how you decided how many impulses to add.

4. In line 135, I think that I would replace “correlation” with “correspondence.”

5. In line 149, I would replace “five impulses are required” with “(7) states that we should use five impulses”.

6. After line 152, refer the reader to the material in lines 406-424 where s/he will see a partial justification for using 5 impulses.

7. In lines 406-424, you added a very informative example. I would like to see one additional case there – an input shaper with six impulses. So far you have justified using more than three or four impulses. It is important to show that six is not a major improvement over five.

Reviewer #3: I have already accepted the paper at the previous revision. I have read it, and in particular, I focused on the responses to the other reviewers' comments. Based on other reviewers' comments, I think the author improved the paper, which is acceptable for publication.

My general comments do not change: I think the results described in this article are predictable, but the paper is well-written and valuable for a reader interested in input shaping methods. Moreover, the idea of designing shapers with constraints on maximal duration is interesting for practical industrial applications that require executing movements in a prescribed duration.

7. PLOS authors have the option to publish the peer review history of their article (what does this mean?). If published, this will include your full peer review and any attached files.

Reviewer #1: No

Reviewer #3: No

---

## [Author Response · Author response to Decision Letter 2]

28 Sep 2022

Please check the attached 'Response to Reviewers_3' file. Thank you.

---

## [Decision Letter · Decision Letter 3]

4 Oct 2022

PONE-D-22-16643R3Specified-duration shapers for suppressing residual vibrationsPLOS ONE

Dear Dr. Kang,

Thank you for submitting your manuscript to PLOS ONE. After careful consideration, we feel that it has merit but does not fully meet PLOS ONE’s publication criteria as it currently stands. Therefore, we invite you to submit a revised version of the manuscript that addresses the points raised during the review process.

We look forward to receiving your revised manuscript.

Kind regards,

Ning Cai, Ph.D.

Academic Editor

PLOS ONE

Reviewers' comments:

Reviewer's Responses to Questions

**Comments to the Author**

1. If the authors have adequately addressed your comments raised in a previous round of review and you feel that this manuscript is now acceptable for publication, you may indicate that here to bypass the “Comments to the Author” section, enter your conflict of interest statement in the “Confidential to Editor” section, and submit your "Accept" recommendation.

Reviewer #1: (No Response)

2. Is the manuscript technically sound, and do the data support the conclusions?

Reviewer #1: Partly

3. Has the statistical analysis been performed appropriately and rigorously? 

Reviewer #1: N/A

4. Have the authors made all data underlying the findings in their manuscript fully available?

Reviewer #1: Yes

5. Is the manuscript presented in an intelligible fashion and written in standard English?

Reviewer #1: Yes

6. Review Comments to the Author

Reviewer #1: I read your revisions with interest. In what follows, the numbering is the same as the numbering in my previous report. As is implied in 1 and 7, I think that “maximal robustness” must be defined clearly.

1. When you speak of maximal robustness, you seem to mean relative to this procedure when a free parameter is added. Generally speaking, maximal robustness would be expected to mean relative to all choices of all relevant parameters.

2. [21 – 26] also include Singer and Singhose among their authors.

3. In my opinion, it is not clear from the text that you are adding impulses in order to add free parameters with which to optimize the design. I still think that this should be pointed out.

4. Thank you.

5. Thank you.

6. Please reword the addition. I recommend, “For an example that provides a partial justification for using…”.

7. I think there is still work to be done. When you add degrees of freedom and optimize, the results should not get worse. If necessary, the optimization should return one amplitude that is set to zero.

I think that we are back to the question of what “maximal robustness” means. If you maximize over all parameter values, adding a new impulse cannot make things worse because forcing one impulse’s amplitude to zero brings us back to the “preceding case.”

If I understand what you are doing correctly, you should be able to fix t_6 – the time of the last pulse – and then optimize by choosing values for A_5,A_6, and t_5 and using the constraints for the five-impulse case to determine the rest of the coefficients. This is still not a search over all possible reasonable value of all the coefficients – so I still would not say that the solution is maximally robust, but it should make sure that your results are not worse than the case with five impulses – as when A_5 is set to zero and t_5 is greater than the value of t_4 in the SD shaper, you should recover the SD shaper. That is, my suggestion is to iterate over possible values of A_5,A_6, and t_5 and use the constraints from the 5-impulse (SD shaper) case to choose the rest of the coefficients.

7. PLOS authors have the option to publish the peer review history of their article (what does this mean?). If published, this will include your full peer review and any attached files.

Reviewer #1: No

---

## [Author Response · Author response to Decision Letter 3]

7 Oct 2022

Response to reviwers letter is attached as a seperate file. Thank you

---

## [Decision Letter · Decision Letter 4]

12 Oct 2022

Specified-duration shapers for suppressing residual vibrations

PONE-D-22-16643R4

Dear Dr. Kang,

We’re pleased to inform you that your manuscript has been judged scientifically suitable for publication and will be formally accepted for publication once it meets all outstanding technical requirements.

Kind regards,

Ning Cai, Ph.D.

Academic Editor

PLOS ONE

Additional Editor Comments (optional):

The review has been completed. Although one reviewer still has some suggestions for improvement, the issues involved are very minor and can be easily addressed in preparing the final version. Since the manuscript has already undergone numerous rounds of revision, in order to save time, I think I'd directly give an acceptation and leave the minor polishing work for the author himself.

Reviewers' comments:

Reviewer's Responses to Questions

**Comments to the Author**

1. If the authors have adequately addressed your comments raised in a previous round of review and you feel that this manuscript is now acceptable for publication, you may indicate that here to bypass the “Comments to the Author” section, enter your conflict of interest statement in the “Confidential to Editor” section, and submit your "Accept" recommendation.

Reviewer #1: (No Response)

Reviewer #5: All comments have been addressed

2. Is the manuscript technically sound, and do the data support the conclusions?

Reviewer #1: Yes

Reviewer #5: Yes

3. Has the statistical analysis been performed appropriately and rigorously? 

Reviewer #1: N/A

Reviewer #5: Yes

4. Have the authors made all data underlying the findings in their manuscript fully available?

Reviewer #1: Yes

Reviewer #5: Yes

5. Is the manuscript presented in an intelligible fashion and written in standard English?

Reviewer #1: Yes

Reviewer #5: Yes

6. Review Comments to the Author

Reviewer #1: I have read your revisions with interest. At this point, I only have some minor comments.

In lines 73 and 74 and all other places where the same correction was made, I think I would use “with the largest” rather than “with the biggest”. Also, I think that I would use “when the magnitude of the last impulse is taken as a free parameter”.

On lines 130 and 131, I think I would say “which can be achieved by adding an impulse and letting its amplitude serve as a free parameter”.

Given that you are no longer saying that the solution is maximal and as your procedure does not really extend to six impulses in the case described in lines 414-443, it may make sense to remove that six-impulse case.

I think that the article should probably be carefully proofread again before being published. (Going over an article carefully several times is always helpful.)

Reviewer #5: Specified-duration shapers are designed in this study, which has some practical value. In this paper, the designing of SD shaper is reasonable and the experimental data is reliable. The subject matter of the paper meets the requirements of the journal.

7. PLOS authors have the option to publish the peer review history of their article (what does this mean?). If published, this will include your full peer review and any attached files.

Reviewer #1: No

Reviewer #5: No

---

## [Editor Report · Acceptance letter]

15 Nov 2022

PONE-D-22-16643R4 

Specified-duration shapers for suppressing residual vibrations 

Dear Dr. Kang:

I'm pleased to inform you that your manuscript has been deemed suitable for publication in PLOS ONE. Congratulations! Your manuscript is now with our production department. 

Kind regards, 

on behalf of

Dr. Ning Cai 

Academic Editor

PLOS ONE